# Bcl-xL blockade targets neutrophils and synergizes with chemotherapy in lung squamous cell carcinoma

Abdullah Mayet [1,2,3], Beatrice Parma[1,2,3], Déborah Lécuyer [1,2,3,4,5], Amaury Defruit [1,2,3], Sarika Rana[1,2,3], Anita Bodac[6], Pieter Demetter[7], Sébastien Denanglaire [1,2], Abbie S Ireland[8], Mariana Brandão [3,9], Thierry Berghmans[9], Fabienne Andris [1,2], Stanislas Goriely[1,2], Jean Yannis Perentes[4,5], Trudy G Oliver [8] & Etienne Meylan [1,2,3,4,5 ✉]

## Abstract

Tumor-associated neutrophils (TANs) represent a large fraction of immune cells in tumors, but how their regulation and function vary in distinct cancer subtypes remains unknown. In $Kras^{LSL-G12D/WT}$; $p53^{fl/fl}$ mouse models of lung adenocarcinoma (LUAD), TANs have an increased lifespan compared to normal neutrophils. Specifically, TANs upregulate the anti-apoptotic protein Bcl-xL, whose blockade by a BH3 mimetic selectively kills ageing TANs and diminishes tumor growth. Here, we have addressed this issue in lung squamous cell carcinoma (LUSC) using the $Rosa26^{LSL-Sox2-IRES-GFP}$; $Nkx2-1^{fl/fl}$; $Lkb1^{fl/fl}$ mouse model, where we demonstrate increased TAN survival with a rise in Bcl-xL similarly to LUAD. However, unlike in LUAD, inhibiting Bcl-xL alone was insufficient to alter tumor progression in LUSC. After carboplatin and paclitaxel treatment, a combination chemotherapy used in human LUSC, we detected increased neutrophils in circulation, spleen and tumors, and increased Bcl-xL in neutrophils and TANs. Bcl-xL blockade decreased the pool of Bcl-xL-high TANs and synergized with chemotherapy. Altogether, our results suggest distinct outcomes for targeting TANs in different tumor types and reinforce the concept of repurposing BH3 mimetics against cancer.

**Keywords** Bcl-xL; Lung Squamous Cell Carcinoma; Mouse Models of Lung Cancer; Tumor-associated Neutrophils
**Subject Categories** Cancer; Immunology; Respiratory System

## Introduction

Neutrophils, the most abundant circulating leucocytes, are critical for the host innate immune defense. However, their impact on cancer development has long been overlooked, which is now changing with recent data obtained from both primary tumors and metastases (Quail et al, 2022). A turning point in our comprehension of tumor-associated neutrophils (TANs) has been tumor-induced polarization, leading to distinct neutrophil states with different, even opposing, functions (Fridlender et al, 2009). Since then, mounting evidence from single-cell analyses has highlighted that TANs exhibit remarkable molecular heterogeneity (Wu et al, 2024; Zilionis et al, 2019), which may translate into diverse capabilities carried by distinct TAN states. In lung cancer, single-cell RNA sequencing analyses of neutrophils from human tumors and murine $Kras^{LSL-G12D/WT}$; $p53^{fl/fl}$ (KP) lung adenocarcinoma (LUAD) have revealed strong conservation between species, and the existence of several molecular states called hN1-hN5 and mN1-mN6, respectively (Zilionis et al, 2019). Of note, the tumor-enriched mN4-mN6 states express SiglecF, a cell surface protein that marks ageing and tumor-supportive TANs (Engblom et al, 2017; Pfirschke et al, 2020). In autochthonous KP mouse models of LUAD (Jackson et al, 2005; Lee et al, 2012), we demonstrated that TAN survival is sustained by increased GLUT1 transporter-mediated glucose uptake and glycolytic metabolism (Ancey et al, 2021). Additionally, we found that expression of the anti-apoptotic Bcl-2 family member, Bcl-xL, increased in TANs compared to normal neutrophils, and continued to do so in ageing TANs that become SiglecF+. Using A-1331852, a highly selective Bcl-xL-inhibitory BH3 mimetic (Leverson et al, 2015; Wang et al, 2020), we selectively eliminated aged and SiglecF+ TANs while preserving young and SiglecF- ones, thereby diminishing KP tumor growth (Bodac et al, 2024). In human non-small cell lung cancer (NSCLC), neutrophils are the most abundant immune cells and are enriched in lung squamous cell carcinoma (LUSC) (Kargl et al, 2017).

[1]Laboratory of Immunobiology, Department of Molecular Biology, Faculty of Sciences, Université Libre de Bruxelles, 6041 Gosselies, Belgium. [2]ULB Center for Research in Immunology (U-CRI), 6041 Gosselies, Belgium. [3]Lung Cancer and Immuno-Oncology Laboratory, Bordet Cancer Research Laboratories, Institut Jules Bordet, Hôpital Universitaire de Bruxelles, Faculty of Medicine, Université Libre de Bruxelles, 1070 Bruxelles, Belgium. [4]Division of Thoracic Surgery, Department of Surgery, Lausanne University Hospital and University of Lausanne, 1011 Lausanne, Switzerland. [5]Agora Cancer Research Center, and Swiss Cancer Center Léman, Lausanne, Switzerland. [6]Swiss Institute for Experimental Cancer Research, School of Life Sciences, Ecole Polytechnique Fédérale de Lausanne, 1015 Lausanne, Switzerland. [7]Cerba Path, Division CMP, 1070 Bruxelles, Belgium. [8]Department of Pharmacology and Cancer Biology, Duke University, Durham, NC 27708, USA. [9]Thoracic Oncology Functional Unit, Institut Jules Bordet, Hôpital Universitaire de Bruxelles, Faculty of Medicine, Université Libre de Bruxelles, 1070 Bruxelles, Belgium. ✉E-mail: Meylan.Etienne@chuv.ch

Correspondingly, genetically engineered mouse models of human LUSC show increased infiltration by neutrophils compared to LUAD models, as exemplified in a comparative analysis between *Lkb1*[fl/fl]; *Pten*[fl/fl] (LUSC) and K or KP tumors (LUAD) (Xu et al, 2014). Loss of tumor suppressor Lkb1 is an important factor driving neutrophil recruitment: when combined with *Kras* mutation, *Lkb1* deletion increased neutrophil infiltration and favored an immunosuppressive environment (Koyama et al, 2016; Nagaraj et al, 2017). In the *Kras*[LSL-G12D/WT]; *Lkb1*[fl/fl] model, tumors tend to transdifferentiate from adeno- to squamous histology during their progression (Han et al, 2014), a process that has been documented in *KRAS* mutant human LUAD upon treatment with mutant selective KRAS inhibitors (Awad et al, 2021). In another model showing a transition and progression from LUAD to LUSC, the *Rosa26*[LSL-Sox2-IRES-GFP]; *Nkx2-1*[fl/fl]; *Lkb1*[fl/fl] (SNL) mouse model, there were higher amounts of TANs compared to KP LUAD (Mollaoglu et al, 2018), while elevated quantities of TANs were also reported in another LUSC model driven by *Sox2* activation combined with *Pten* and *Cdkn2ab* deletion (Ferone et al, 2016). Single-cell RNA sequencing of TANs and circulating neutrophils from SNL mice revealed the presence of a SiglecF[+] TAN subset, like in KP tumors, again suggesting an increased lifespan compared to normal neutrophils. Neutrophil depletion using an anti-Ly6G antibody for three weeks reduced the number of squamous lesions, suggesting TANs are implicated in the transition from LUAD or in the development or survival of LUSC (Mollaoglu et al, 2018). In the present study, we aimed to evaluate the response to Bcl-xL blockade in TANs from the SNL tumor model, when applied as single agent or in combination with therapies used to combat human LUSC.

# Results

## Neutrophils constitute the main fraction of tumor-infiltrating immune cells and are long-lived in the SNL mouse model

Because TANs, more precisely the aged, Bcl-xL[high], SiglecF[high] subset, support tumors in KP models of LUAD (Engblom et al, 2017; Pfirschke et al, 2020; Ancey et al, 2021; Bodac et al, 2024; Faget et al, 2017), we wanted to know whether this was similar or different in LUSC, the other main histological NSCLC subtype. First, in the adenoviral vector Ad5CMVCre-driven SNL mouse model we measured an increased number of TANs in LUSC compared to LUAD lesions (Fig. EV1A). Next, we analyzed and compared SNL to KP tumors using 16-color flow cytometry (Appendix Fig. S1), monitoring the principal cell types of the tumor immune microenvironment. Confirming previous findings (Mollaoglu et al, 2018), TANs were the most prevalent immune cell population in SNLs, reaching 57.5% ± 34.0% of all immune (CD45[+]) cells compared to 30.7% ± 6.5% in KP tumors (Fig. 1A). Another notable difference between SNL and KP lesions was the increased proportion of T and B lymphocytes in SNL (Fig. 1A). Next, focusing on TANs, we identified varied SiglecF expression levels (Fig. 1B). Similar to KPs, SiglecF expression correlated with that of the anti-apoptotic Bcl-xL protein, increasing from SiglecF low to intermediary to high, suggesting the existence of long-lived TANs in LUSC (Fig. 1C). To directly test this, we used a BrdU

tracking experiment, injecting mice with a single dose of BrdU 6.5 days prior to sacrifice. Due to their short half-life, at this time point most neutrophils in circulation or in tissues are BrdU[-], because BrdU[+] neutrophils (whose proliferative progenitors in the bone marrow had incorporated the DNA intercalating agent) have died and been replaced by negative cells (Ancey et al, 2021; Boivin et al, 2021). Accordingly, >90% of lung or splenic neutrophils from healthy or tumor-bearing mice were BrdU[-] one week after injection. In contrast, 27.3% ± 7.8% of TANs were BrdU[+] (Fig. 1D), demonstrating an increase in the lifespan of these cells in SNL tumors, similar to what we reported in KP tumors (Ancey et al, 2021). The SiglecF[+] fraction (comprising both intermediary and high) was enriched in old TANs compared to its SiglecF[low] counterpart, since 57.8% ± 4.9% *versus* 40.3% ± 4.9% cells were BrdU[+] (Fig. 1E). By comparing SNL and KP TANs, we measured a very significant difference in the neutrophil maturation-associated marker, CD101. While only 10.3% ± 4.6% of KP TANs were CD101 negative, they accounted for 29.2% ± 7.9% of SNL TANs, suggesting an increased preponderance of immature neutrophils in SNL tumors (Fig. EV1B). This was in line with a significantly higher expression of the lipopolysaccharide co-receptor and activation marker CD14 in KP compared to SNL TANs (68.1% ± 6% versus 46.1% ± 20.2%). Finally, PD-L1[hi] TANs, already the main fraction of KP TANs (79.9% ± 4.7%), comprised almost all SNL TANs (92.9% ± 3.2%). These data suggest phenotypic differences in TANs from KP mice and those from SNL mice, at least when considered as a whole. To more globally assess the variability of TANs between the two models, we applied the gene signatures of the KP mN1-mN6 molecular states (Data ref: Zilionis et al, 2019) to the single-cell RNA sequencing data of peripheral blood neutrophils (PBN) and TANs from SNLs (Data ref: Mollaoglu et al, 2018). In KPs, mN1 and mN2 are 10-fold enriched in healthy lung neutrophils compared to TANs; mN3 and mN6 are exclusively present in tumor tissue, while mN4 and mN5 are 10–20-fold enriched in tumor tissue (Zilionis et al, 2019). Consistent with this, we identified an enrichment of mN3 and mN5 in SNL TANs compared to PBNs, and an enrichment of mN1 and mN2 in PBNs compared to TANs (Fig. EV1C,D). For mN4 and for mN6 showing a bi-modal distribution, it was harder to conclude. Thus, these data highlight similarities but also variation between SNL and KP TANs, and demonstrate prolonged survival of SNL TANs, which constitute the major fraction of immune cells infiltrating SNL tumors.

## Bcl-xL inhibition does not diminish SNL tumor growth

In the KP model, Bcl-xL blockade using a highly selective inhibitor, A-1331582, kills old, SiglecF[+] TANs while preserving younger neutrophils that are SiglecF[-], and delays tumor growth when applied as single agent (Bodac et al, 2024). To determine whether SNL tumors are also sensitive to Bcl-xL inhibition, we initially applied the same protocol, treating mice with an intermittent regimen (5 days ON, 2 days OFF, Fig. EV2A). A-1331852 efficiently suppressed the ageing of TANs, as it diminished the proportions of BrdU[+] and SiglecF[int/hi] TANs at 6.5 days after BrdU injection, while augmenting SiglecF[low] TANs (Fig. EV2B,C). However, no reduction in tumor growth was detected after Bcl-xL blockade, as monitored by in vivo longitudinal micro-computed tomography (µCT) (Fig. EV2D). This treatment dose was probably too high in SNLs,

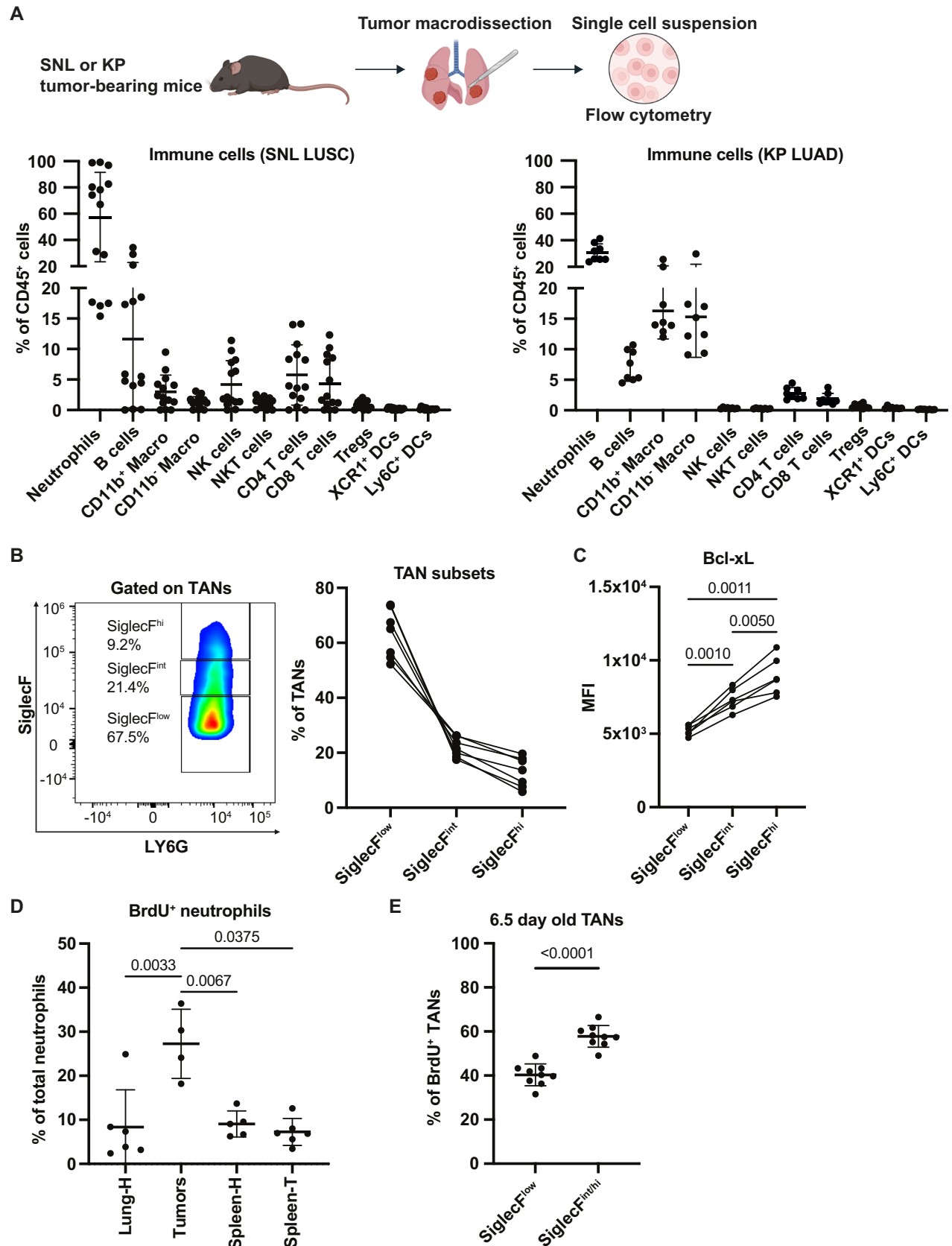

◀  **Figure 1. SNL tumors are strongly infiltrated by long-lived neutrophils.**

(A) Immune cell proportions in SNL (left) vs KP (right) tumors ($n = 14$ tumors for SNL, $n = 8$ tumors for KP). (B) (Left) Pseudocolor plot showing an example of SiglecF expression in SNL TANs. (Right) Proportions of SiglecF TANs (low, intermediary or high) among total TANs in SNL tumors ($n = 7$ tumors). (C) MFI of Bcl-xL expression in SiglecF TAN subsets quantified by flow cytometry ($n = 6$ tumors). (D) 6.5 days old BrdU[+] neutrophils in the indicated tissues from healthy (H) or tumor-bearing (T) mice among total neutrophils ($n = 4$–6 samples for each tissue). (E) 6.5 days old BrdU[+] SiglecF TANs among total TANs ($n = 9$ tumors). Data information: p-value was determined using one-way ANOVA for (C) and (D). For (E) paired t-test was used. For (A), (D) and (E), data are shown as mean ± SD.

because it very significantly decreased (i) the proportion of all TANs to 12.7% ± 1.9% of tumor-infiltrating immune cells (Fig. EV2E), which could include cells with anti-tumor functions, and (ii) the proportion of CD14[+] TANs, a subset known to increase with successful neutrophil-dependent anti-tumoral anti-CD40 or anti-OX40 immunotherapies (Hirschhorn et al, 2023; Gungabeesoon et al, 2023) (Fig. EV2F). These data prompted us to adapt our protocol by diminishing the total dose applied, treating mice every other day for two- or three weeks (Fig. 2A). After two weeks of treatment, A-1331852 significantly diminished the proportion of SiglecF[int/hi] TANs and increased that of SiglecF[low] TANs, while total TANs and CD14[+] TANs were preserved (Fig. 2B–D). However, treatment with A-1331852 alone did not affect tumor growth (Fig. 2E,F). After three weeks of treatment, no change was observed in the proportions of TANs or SiglecF TAN subsets (Fig. 2G,H), while Bcl-xL protein expression in TANs was very significantly reduced (Fig. 2I). Specifically, after two weeks of treatment, a decrease in Bcl-xL was observed only for SiglecF[int/hi] TANs, which became significant for both SiglecF[low] and SiglecF[int/hi] subsets one week later (Fig. EV2G,H). As there was no change in total TANs, these results suggest that long-term blockade of Bcl-xL replaces Bcl-xL[high] with Bcl-xL[low] TANs. Thus, Bcl-xL[high] TANs are vulnerable to A-1331852 treatment, which is not sufficient to diminish SNL tumor growth.

## SNL tumors are resistant to anti-PD-1 + Bcl-xL blockade and to standard chemotherapies

Although no tumor growth delay was observed in response to A-1331852 and given that TANs from this model are positive for the immune checkpoint PD-L1, surprisingly expressing it at higher levels in SiglecF[low] and SiglecF[int] compared to SiglecF[hi] TANs (Fig. EV3A), we next decided to test whether Bcl-xL inhibition could effectively combine with anti-PD-1 immunotherapy. Although PD-L1 protein expression was sustained in TANs in response to two- or three-week A-1331852 treatment (Fig. EV3B–D), tumor growth rate analyses by μCT failed to reveal any change between SNL mice treated with A-1331852 + anti-PD-1 and vehicle-treated animals (Fig. EV3E,F). After two weeks of treatment, anti-PD-1 alone did not change the SiglecF or Bcl-xL ratios in TANs and did not modify the effect of A-1331852 on these ratios (Fig. EV3G–I). Among all tested tumor-infiltrating immune cells, anti-PD-1 only significantly decreased the proportion of TANs (Appendix Fig. S2).

Because chemotherapy is also a treatment option for NSCLC, we next decided to evaluate the effect of combination chemotherapy (carboplatin and paclitaxel) on SNL tumors, which is often prescribed for patients with LUSC. After a cycle of treatment consisting of three weekly injections following a protocol previously used in KP mice (Pfirschke et al, 2016) (Fig. 3A), we did not detect a decrease in tumor growth rate (Fig. 3B,C). To determine how chemotherapy may affect the tumor immune microenvironment, we used our multi-color flow cytometry approach (see Fig. 1A;

Appendix Fig. S1) to measure the tumor-infiltrating immune cells in control versus treated tumors. Among the immune cells examined, we did not observe major changes in their proportion except for a trend toward reduced B cells in treated tumors. Neutrophils were an exception, as their abundance increased very significantly in tumors treated with chemotherapy, reaching 74.2% ± 22.1% (Fig. 3D). This rise could not be attributed to a specific TAN subset based on SiglecF expression (Fig. 3E). In TANs, but not in tumor cells, Bcl-xL expression measured by flow cytometry was higher after chemotherapy treatment (Fig. 3F,G), which, surprisingly, was only significant for the SiglecF[low] TAN subset (Fig. 3H). Interestingly, TANs from chemotherapy-treated SNL mice were similar to TANs from untreated KP mice in terms of Bcl-xL expression (Fig. EV4A). The neutrophil response to chemotherapy was not limited to the tumor but occurred systemically, as evidenced by increased proportion and number of circulating neutrophils, significantly higher expression of Bcl-xL by circulating neutrophils, and increased abundance of splenic neutrophils tending toward higher Bcl-xL levels (Fig. EV4B–F). This systemic response, which could be reactive granulopoiesis induced by chemotherapy, was accompanied by changes in stem and progenitor cells in the bone marrow and spleen. Specifically, among long- (LT) and short- (ST) term hematopoietic stem cells (HSC), common myeloid progenitor (CMP), granulocyte-monocyte progenitor (GMP) cells, and multipotent progenitor ((MPP)-4, a lymphoid-primed progenitor), we quantified a significant decrease of CMPs and GMPs in the bone marrow, and CMPs in the spleen (Fig. EV4G,H; Appendix Fig. S3). These cells may be extensively used and consumed under the pressure imposed by chemotherapy to sustain neutrophil production, or be sensitive to chemotherapy. Specifically in the spleen, there was a significant increase of LT-HSCs, which could contribute to maintaining or even increasing neutrophil production in this extramedullary site in response to chemotherapy (Fig. EV4H). Thus, combined carboplatin + paclitaxel therapy is relatively inefficient in SNL mice and triggers a systemic neutrophil response leading to the accumulation of TANs strongly expressing Bcl-xL.

## Bcl-xL blockade synergizes with chemotherapy

Since tumor infiltration by neutrophils can be associated with a lack of response to chemotherapy (Jaillon et al, 2020), and since Bcl-xL[high] TANs are selectively eliminated by A-1331852 in both KP (Bodac et al, 2024) and SNL tumors (see Fig. 2I), we next decided to investigate whether increased Bcl-xL expression in TANs after chemotherapy might offer an opportunity for combination therapy. We therefore treated SNL tumor-bearing mice with A-1331852 combined with carboplatin + paclitaxel, using the scheme previously established for separate treatments (Fig. 4A). Bcl-xL blockade synergized with chemotherapy, as measured by μCT imaging, which demonstrated a reduction in tumor growth rate 2.6 weeks after the initiation of treatment, with even some tumors

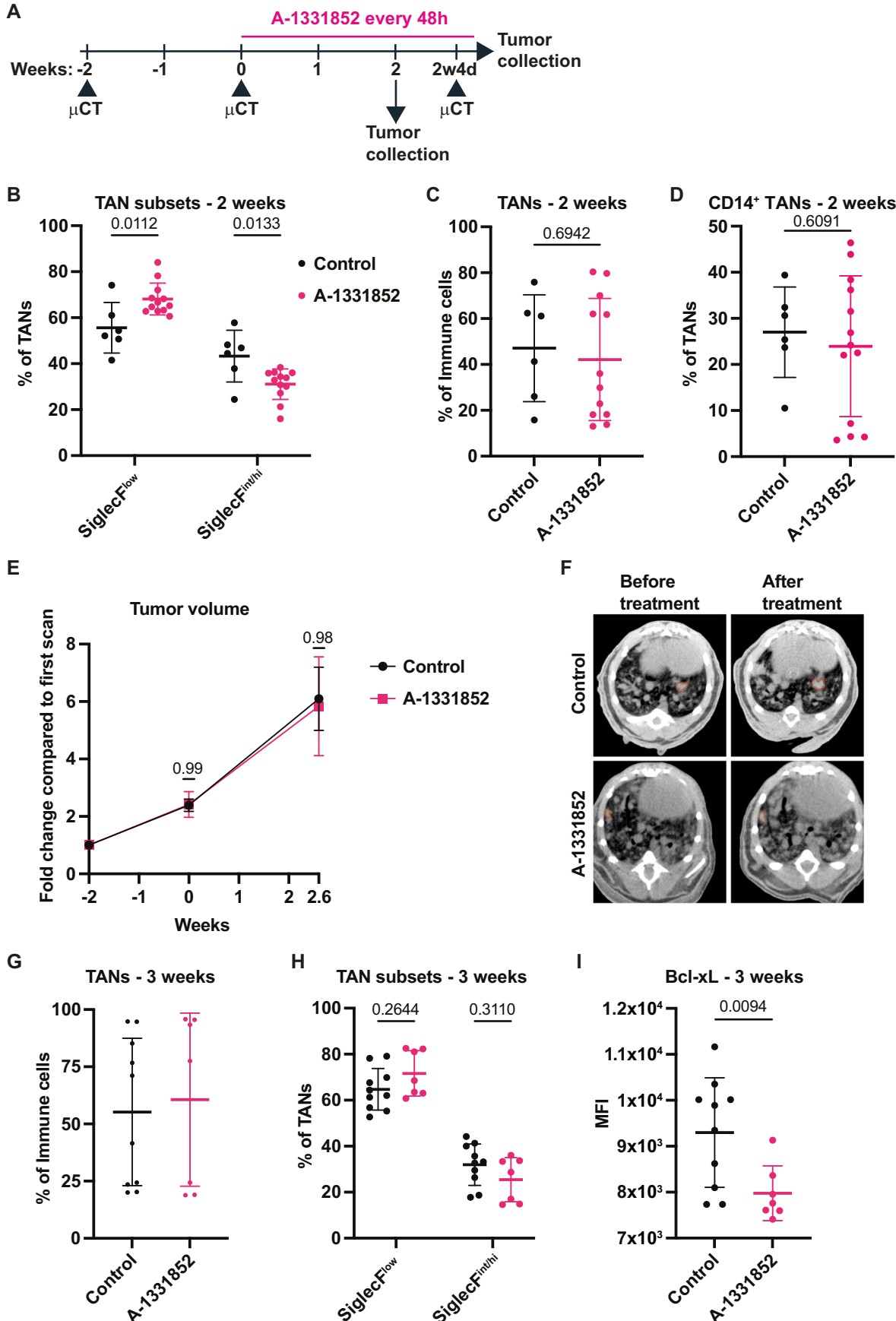

**Figure 2. Bcl-xL targeting eliminates Bcl-xL^high TANs but does not impact SNL tumor growth.**

(A) Scheme of in vivo Bcl-xL blockade regimen. SNL mice were treated for either two or three weeks. (B) Proportions of SiglecF TAN subsets among total TANs ($n = 6$ tumors for control and $n = 12$ tumors for treated group). (C) Proportions of TANs among all immune cells ($n = 6$ tumors for control and $n = 12$ for treated group). (D) Proportions of CD14$^+$ TANs among total TANs ($n = 6$ tumors for control and $n = 13$ tumors for treated group). (E) Tumor growth kinetics measured by μCT ($n = 13$ tumors for control, $n = 4$ tumors treated group). (F) Representative images showing tumor growth measured at the start of the treatment and 2.6 weeks later. (G) Proportions of TANs among all immune cells ($n = 10$ tumors for control and $n = 7$ for treated group). (H) Proportions of SiglecF TAN subsets among total TANs ($n = 10$ tumors for control and $n = 7$ tumors for treated group). (I) MFI of Bcl-xL expression in TANs ($n = 10$ tumors for control and $n = 7$ tumors for treated group). Data information: p-value was determined using 2-way ANOVA for (B), (E) and (H). For (C), (D) and (I) Welch's t-test was used. Data are shown as mean ± SD except for (E), which is shown as mean ± SEM.

regressing (Fig. 4B,C). To interrogate how the drugs might impact TANs, we measured their abundance using flow cytometry. First, we noticed that the chemotherapy-induced increase in the proportion of TANs (see Fig. 3D) was attenuated upon Bcl-xL blockade (Fig. 4D). In fact, combined treatment reduced the total number of TANs (Fig. 4E), without changing the ratio of TANs differentially expressing SiglecF (Fig. 4F). However, in each of the three SiglecF TAN subsets, but not in peripheral blood neutrophils, we measured a very significant decrease in Bcl-xL protein expression in the treated conditions (Fig. 4G). Moreover, TANs expressing intermediate to high Bcl-xL levels (Bcl-xL^int/hi) almost disappeared, their frequency dropping from 25.6% ± 7.5% to only 5.4% ± 3.4% (Fig. 4H). To determine whether blocking Bcl-xL improves the efficacy of chemotherapy by acting on TANs, we hypothesized that depleting neutrophils could recapitulate the effect of A-1331852. Partial neutrophil depletion using a dual antibody strategy (rat anti-Ly6G + anti-rat secondary antibody) (Boivin et al, 2020) improved the efficacy of chemotherapy to a similar extent as A-1331852 (Fig. EV5A,B), supporting a scenario in which Bcl-xL blockade in TANs affects the response to chemotherapy. However, it should be noted that in vitro, A-1331852 treatment enhanced the sensitivity of two human squamous tumor cell lines to carboplatin and paclitaxel (Appendix Fig. S4). We do not currently know whether and how tumor cells from the SNL model respond to chemotherapy when Bcl-xL is blocked. Thus, blocking Bcl-xL efficiently sensitizes SNL tumors to chemotherapy and eliminates TANs expressing elevated levels of Bcl-xL.

## Discussion

The role of innate immune cells including neutrophils in cancer immunology represents an important yet still underexplored area of investigation. As the heterogeneity of TANs is increasingly recognized, resulting in divergent roles for specific TAN subsets, it is important to identify to which extent these cells have similar or different functions and vulnerabilities in distinct cancer types, stages of progression, or in response to successful or failed therapies. Complementing our previous study (Bodac et al, 2024), we have here provided evidence for shared and exclusive attributes between TANs from two molecularly and histologically distinct mouse models of NSCLC. In both KP and SNL tumors, ageing TANs increasingly express SiglecF and Bcl-xL, and blocking Bcl-xL retards the growth of KP tumors—at least partially via TAN targeting—while SNL tumors remain unresponsive. In two recent studies, successful anti-CD40 or anti-OX40 therapies in mouse models of lung or colorectal cancer (CD40) or melanoma (OX40) implied the expansion of a pool of anti-tumor TANs otherwise not predominant in untreated tumors. These therapy-associated TANs highly expressed multiple neutrophil activation markers, including interferon-stimulated genes (ISG) and the toll-like receptor 4 co-receptor, CD14 (Hirschhorn et al, 2023; Gungabeesoon et al, 2023).

In contrast, we find that blocking Bcl-xL with five times per week A-1331852 treatment in SNLs reduced the abundance of CD14$^+$ TANs. For a successful tumor response to neutrophil-mediated therapies, it may therefore be crucial to identify ways of maintaining or increasing the abundance of one or more specific TAN subsets, for example those expressing CD14 and certain ISGs.

While the anti-apoptotic Bcl-2 family has long been considered for cancer cell killing, only one BH3 mimetic, venetoclax that targets Bcl2, has obtained clinical approval for the treatment of hematological malignancies, including chronic lymphocytic leukemia, acute myeloid leukemia and multiple myeloma (Lasica and Anderson, 2021). Sonrotoclax, a second-generation Bcl2 inhibitor, is currently under clinical evaluation (Guo et al, 2024; Liu et al, 2024). The development of BH3 mimetics targeting other members of the anti-apoptotic Bcl-2 family has been hampered by important side effects. Indeed, dual Bcl-2/Bcl-xL inhibitors (ABT-737 or navitoclax) induce thrombocytopenia, as platelet survival depends on Bcl-xL (Mason et al, 2007; Zhang et al, 2007). Furthermore, small-molecule inhibition of Bcl-xL leads to rapid, on-target fatal cardiovascular toxicity in dogs, which can be alleviated—along with thrombocytopenia—by the development of antibody-drug conjugates (Judd et al, 2024). Such approach can efficiently deliver payloads to the tumor site, which may release and target Bcl-xL inhibitory molecules to both tumor cells and TANs, providing a promising approach for clinical use.

Our results demonstrate that SNL tumors are relatively resistant to combination chemotherapy, and to anti-PD-1 immunotherapy with A-1331852, including therapies used in the clinic. This is important, because current treatment options are rarely curative in advanced disease, and usually act transiently, if at all. Hence, the SNL model could be the subject of further preclinical investigation, to identify the most effective therapy combinations, and clinically applicable ways for sensitizing tumors to chemotherapies or immunotherapies.

A common side effect of the use of chemotherapy in cancer patients is the development of neutropenia, which can impair the body's innate immune response to infections. This can be compensated by reactive granulopoiesis, i.e., enhanced granulopoiesis in the absence of microbial infection, enabling to recover functional neutrophils thereby reducing risks of severe infections (Manz and Boettcher, 2014). As we found an increased proportion and number of neutrophils in blood, spleen and tumors of chemotherapy-treated animals, we favor the possibility that this response models chemotherapy-induced reactive granulopoiesis. Based on Bcl-xL analysis alone, which shows higher protein expression after treatment particularly in the SiglecF^low TAN subset, we anticipate chemotherapy-induced TANs are phenotypically different from TANs in untreated conditions. For this clinically relevant situation, we suggest that SNL tumor-bearing mice could be used to interrogate neutrophil and TAN phenotypes during reactive granulopoiesis.

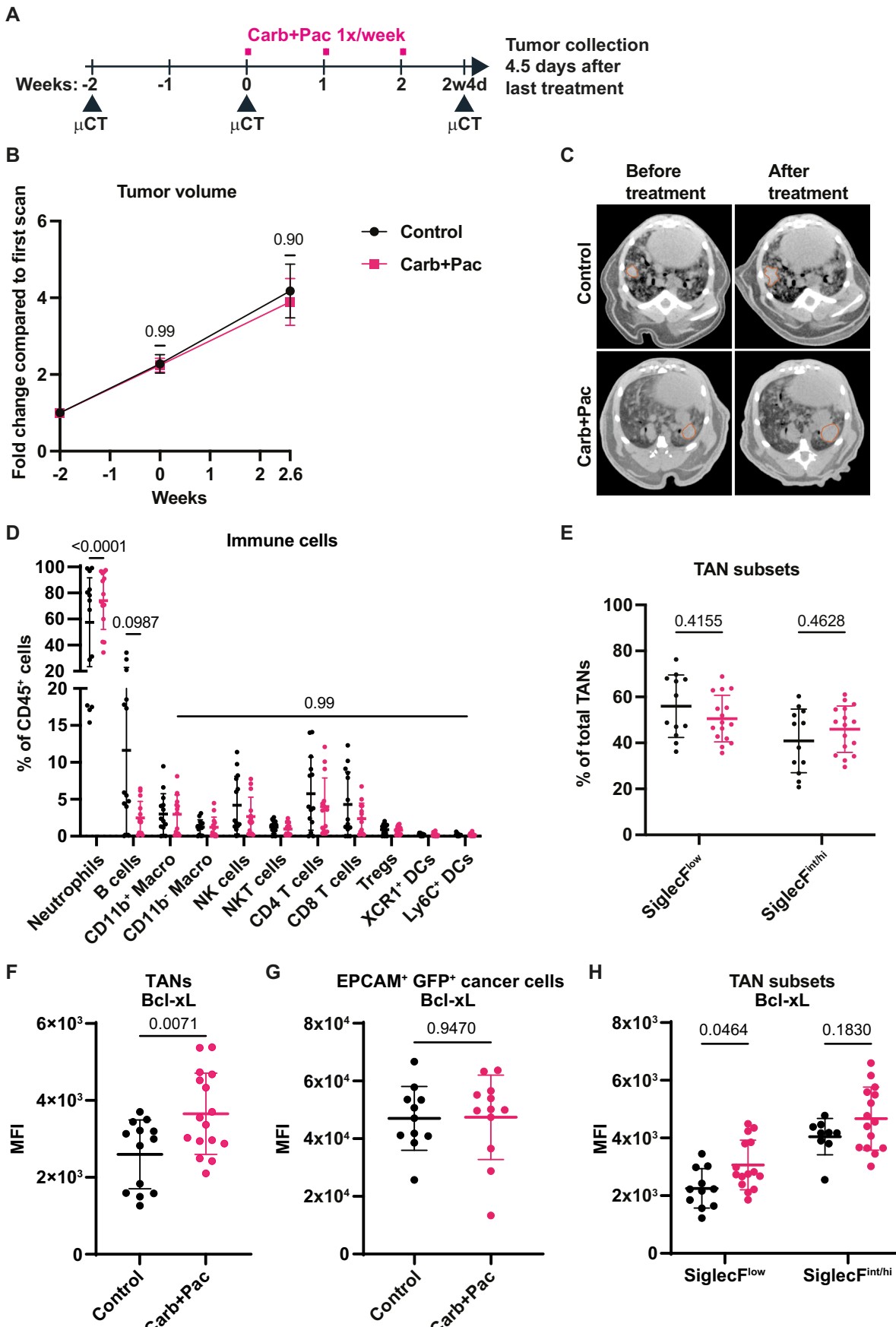

**Figure 3. Chemotherapy further increases Bcl-xL expression in TANs but does not impede tumor growth.**

(A) Scheme of in vivo double chemotherapy regimen. (B) Tumor growth kinetics measured by μCT ($n = 8$ tumors for control and $n = 10$ tumors for treated group). (C) Representative images showing tumor growth measured at the start of the treatment and 2.6 weeks later. (D) Proportions of the main immune cell populations ($n = 14$ tumors for both groups). (E) Proportions of SiglecF TAN subsets among total TANs ($n = 12$ tumors for control and $n = 16$ for treated group). (F) MFI of Bcl-xL expression in TANs ($n = 13$ tumors for control and $n = 16$ for treated group). (G) MFI of Bcl-xL expression in EPCAM$^+$GFP$^+$ SNL cancer cells ($n = 11$ tumors for control and $n = 12$ for treated group). (H) MFI of Bcl-xL expression within SiglecF$^{low}$ ($n = 11$ tumors for control and $n = 15$ for treated group) and SiglecF$^{int/hi}$ ($n = 9$ for control and $n = 15$ for treated group) TANs. Data information: $p$-value was determined using 2-way ANOVA for (B), (D), (E), and (H). For (F), Welch's t-test was used and for (G), unpaired t-test was used. Data are shown as mean ± SD except for (B), which is shown as mean ± SEM.

Chemoresistance can be driven by the tumor microenvironment. Here, we provide evidence that SNL tumors are sensitized to combined carboplatin and paclitaxel when A-1331852 was added to the treatment protocol. Our current interpretation is that chemotherapy induces systemic and local changes that enhance Bcl-xL expression in neutrophils and TANs, rendering them vulnerable to Bcl-xL blockade. After selective removal of Bcl-xL$^{high}$ TANs, SNL tumors become sensitive to chemotherapy, providing therapeutic synergy. This is compatible with recent data linking chemoresistance to neutrophil function. In a mouse model of experimental breast cancer-to-lung metastasis, neutrophil extracellular traps (NET) were induced in response to chemotherapy in the lung environment, triggering TGF-β activation and cancer cell epithelial-mesenchymal transition, leading to chemoresistance and increased metastatic burden (Mousset et al, 2023). Furthermore, NETs produced upon local lung inflammation (triggered by lipopolysaccharide or tobacco smoke) or systemic chemotherapy, were shown to awaken lung-disseminated cancer cells from dormancy, thereby enhancing the development of lung metastases (He et al, 2025; Albrengues et al, 2018). In neoadjuvant chemotherapy-treated human breast cancers, chemoresistant tumors contained, in comparison to chemosensitive samples, abundant ferroptotic TANs, which were characterized by altered phospholipid profiles with decreased monounsaturated and increased polyunsaturated fatty acids. These chemoresistance-associated ferroptotic TANs secreted high quantities of PGE$_2$ and IDO and were also immune suppressive (Zeng et al, 2025). In future research, it will be interesting to determine whether there are functional links between Bcl-xL activity, NET formation and release, and ferroptosis induction in the response of primary lung tumors to current therapies. In conclusion, our work established that SNL tumors are relatively resistant to multiple treatment modalities used clinically. Combination chemotherapy triggered reactive granulopoiesis marked by elevated Bcl-xL expression in circulating neutrophils and TANs, resulting in sensitization to Bcl-xL blockade and synergy with chemotherapy. This work may pave the way for the identification of new clinically relevant approaches to eliminating tumor-supportive neutrophils as part of combination therapies.

## Methods

### Reagents and tools table

| Reagent/ Resource | Reference or Source | Identifier or Catalog Number |
| --- | --- | --- |
| **Experimental models** | | |
| Rosa26$^{LSL-Sox2-IRES-GFP}$; Nkx2-1$^{fl/fl}$; Lkb1$^{fl/fl}$ (M. musculus) | Mollaoglu et al (2018) (Immunity) | Prof. Trudy G Oliver, Duke University, USA |
| Kras$^{LSL-G12D/WT}$; p53$^{Flox/Flox}$ (M. musculus) | The Jackson Laboratory | RRID:IMSR_JAX:008179 RRID:IMSR_JAX:008462 |
| FaDu cells (H. sapiens) | ATCC | HTB-43 |
| SK-MES-1 cells (H. sapiens) | ATCC | HTB-58 |
| **Recombinant DNA** | | |
| Ad5CMVCre viral vector | University of Iowa Viral Vector Core Facility | VVC-U of Iowa-5 |
| Ad5mSPC-Cre viral vector | University of Iowa Viral Vector Core Facility | VVC-Berns-1168 |
| **Antibodies** | | |
| Rat Isotype | Bio X Cell | BE0089 |
| Mouse Isotype | Bio X Cell | BE0085 |
| Rat anti-LY6G | Bio X Cell | BP0075-1 |
| Mouse anti-Rat Kappa | Bio X Cell | BE0122 |
| Rat anti-PD-1 | Bio X Cell | BE0273 |
| Rat anti-B220 BV480 | BD Biosciences | 565631 |
| Rabbit anti-Bcl-xL PE | Abcam | AB208747 |
| Mouse anti-BrdU APC | BioLegend | 339808 |
| Mouse anti-CD11b AF700 | BD Biosciences | 557960 |
| Rat anti-CD11b BV711 | BD Biosciences | 563168 |
| Hamster anti-CD11c PE-CY5.5 | Invitrogen | 35-0114-82 |
| Rat anti-CD14 PECY7 | BioLegend | 123316 |
| Rat anti-CD274 BV786 | BioLegend | 124331 |
| Hamster anti-CD3 BV711 | BD Biosciences | 563123 |
| Rat anti-CD4 BUV496 | BD Biosciences | 612952 |
| Rat anti-CD45 BUV395 | BD Biosciences | 564279 |

| Reagent/Resource | Reference or Source | Identifier or Catalog Number |
|---|---|---|
| Rat anti-CD45 PERCP-CY5.5 | BD Biosciences | 561869 |
| Rat anti-CD8 APC-H7 | BD Biosciences | 560182 |
| Rat anti-F4/80 BV421 | BD Biosciences | 565411 |
| Rat anti-F4/80 PECF594 | BD Biosciences | 565613 |
| Rat anti-FOXP3 PE-CY5 | Invitrogen | 15-5773-82 |
| Rat anti-Ly-6C PERCP-CY5.5 | Invitrogen | 45-5932-80 |
| Rat anti-Ly-6G FITC | BD Biosciences | 551460 |
| Rat anti-Ly-6G RB613 | BD Biosciences | 759159 |
| Rat anti-MHC-II BUV737 | BD Biosciences | 569176 |
| Goat anti-MPO | R&D Systems | AF3667 |
| Mouse anti-NK1.1 BV421 | BioLegend | 108741 |
| Rat anti-SiglecF BB515 | BD Biosciences | 564514 |
| Rat anti-SiglecF PE | BD Biosciences | 552126 |
| Mouse anti-XCR1 BV650 | BioLegend | 148220 |
| Hamster anti-CD101 APC | Invitrogen | 17-0114-82 |
| Rat anti-EPCAM PE-CY7 | Invitrogen | 25-5791-80 |
| Mouse anti-GFP AF647 | BD Biosciences | 565197 |
| Rat anti-TER119 APC | BioLegend | 116211 |
| Rat anti-B220 APC | BD Biosciences | 553092 |
| Hamster anti-CD3 APC | BioLegend | 100311 |
| Rat anti-CD4 APC | BD Biosciences | 553051 |
| Rat anti-CD8 APC | BD Biosciences | 553035 |
| Rat anti-CD11b APC | Invitrogen | 17-0112-82 |
| Rat anti-Gr1 APC | BioLegend | 108411 |
| Rat anti-c-Kit PE | Invitrogen | 12-1171-81 |
| Rat anti-CD135 PE-CY5 | BioLegend | 135311 |
| Rat anti-Sca-1 BV510 | BioLegend | 108129 |
| Rat anti-CD34 BV421 | BioLegend | 152207 |
| Hamster anti-CD48 BV785 | BioLegend | 103449 |
| Rat anti-CD150 PECF594 | BioLegend | 115935 |

| Reagent/Resource | Reference or Source | Identifier or Catalog Number |
|---|---|---|
| **Oligonucleotides and other sequence-based reagents** | | |
| **Chemicals, Enzymes and other reagents** | | |
| A-1331852 | MedChemExpress | HY-19741 |
| Carboplatin | MedChemExpress | HY-17393 |
| Paclitaxel | MedChemExpress | HY-B0015 |
| PEG300 | MedChemExpress | HY-Y0873 |
| Tween-80 | MedChemExpress | HY-Y1891 |
| iFluor® 860 maleimide | AAT Bioquest | 1408_1mg |
| LIVE/DEAD™ Fixable Near-IR Dead Cell Stain Kit | Invitrogen | L34976 |
| BrdU | MedChemExpress | HY-15910 |
| DNAse I | Merck | D4513 |
| Collagenase | Merck | C2675 |
| HBSS | ThermoFisher Scientific | 14025092 |
| MEM | ThermoFisher Scientific | 11095080 |
| **Software** | | |
| NRecon | https://www.microphotonics.com/micro-ct-systems/nrecon-reconstruction-software/ | |
| FlowJo | https://www.flowjo.com/ | |
| QuPath | https://qupath.github.io/ | |
| GraphPad Prism | https://www.graphpad.com/ | |
| bioRender | https://www.biorender.com/ | |
| Horos | https://horosproject.org | |
| **Other** | | |
| Bruker Skyscan 1278 microCT | Bruker | |
| GentleMACS tissue dissociator | Miltenyi | |
| PT Link | Agilent | |
| Hamamatsu NanoZoomer S360MD slide scanner | Hamamatsu | |
| Beckman Coulter CytoFLEX LX | Beckman Coulter | |
| BD LSRFortessa | BD Biosciences | |
| Incucyte® SX5 | Sartorius | |

## Methods and protocols

### Mouse models

*Rosa26*<sup>LSL-Sox2-IRES-GFP</sup>; *Nkx2-1*<sup>fl/fl</sup>; *Lkb1*<sup>fl/fl</sup> (SNL) mice were a kind gift of Prof. T.G. Oliver (Duke University). At the age of 12–18 weeks, mice were intratracheally instilled with $0.25{-}1 \times 10^8$ pfu Ad5CMVCre viral vector (purchased from University of Iowa Viral Vector Core facility) to generate LUSC tumors. For LUAD

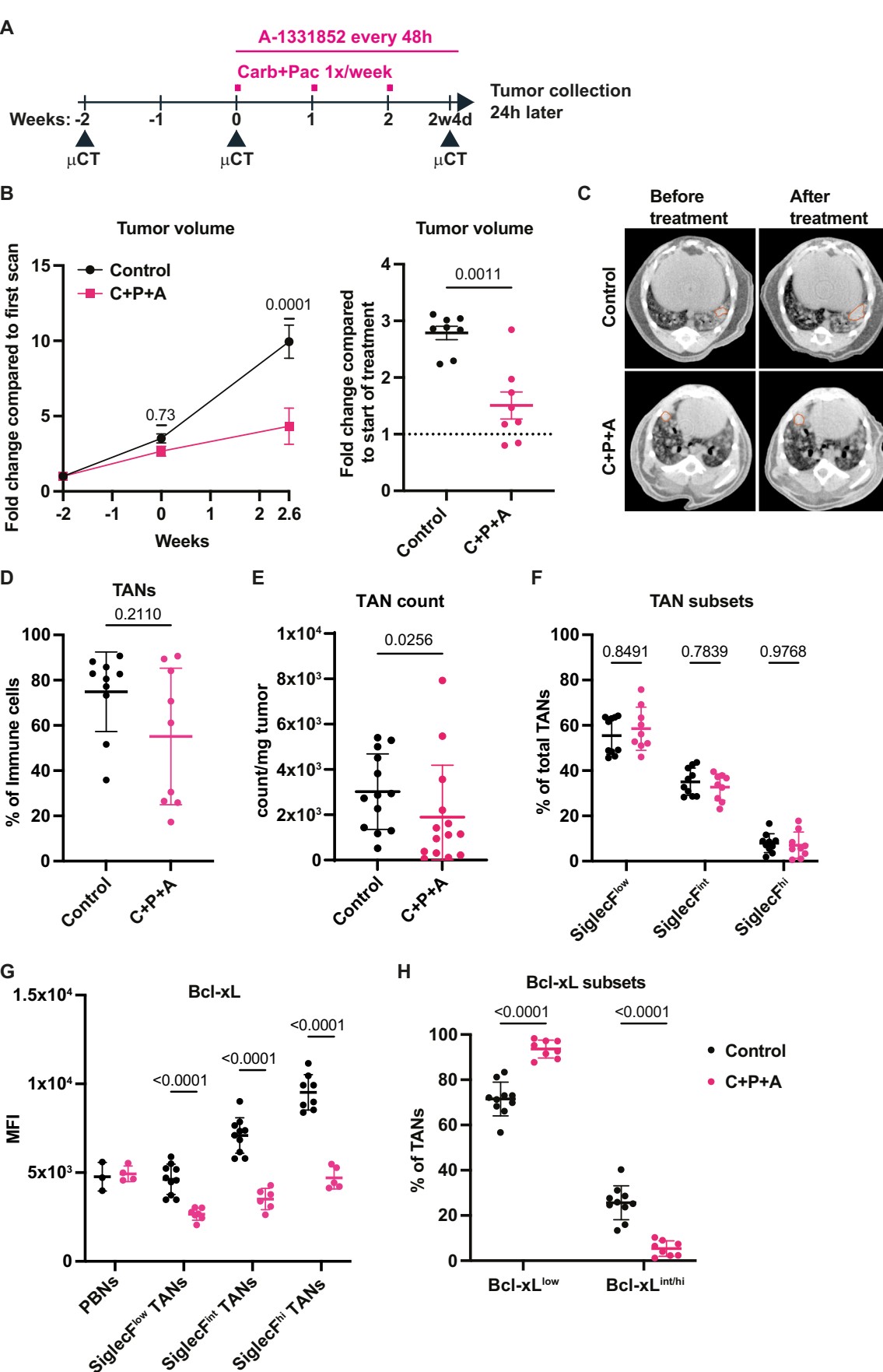

◀ **Figure 4. Bcl-xL blockade synergizes with chemotherapy.**

(A) Scheme of in vivo combination treatment regimen. (B) Tumor growth kinetics measured by μCT ($n = 8$ tumors for both the control and the carboplatin + paclitaxel + A-1331852 (C + P + A) treated groups). (C) Representative images showing tumor growth measured at the start of the treatment and 2.6 weeks later. (D) Proportions of TANs among all immune cells ($n = 10$ tumors for control and $n = 9$ for treated group). (E) TAN count per mg of tumor quantified via flow cytometry ($n = 13$ tumors for control and $n = 14$ for treated group). (F) Proportions of SiglecF TAN subsets among total TANs ($n = 10$ tumors for control and $n = 9$ for treated group). (G) MFI of Bcl-xL expression in PBNs and TANs ($n = 3$–10 for control group and $n = 4$–7 for treated group). (H) Proportions of TANs expressing low (Bcl-xL$^{low}$) or intermediate-to-high (Bcl-xL$^{int/hi}$) Bcl-xL levels among total TANs ($n = 10$ tumors for control and $n = 8$ for treated group). Data information: $p$-value was determined using 2-way ANOVA for (B left), (F), (G) and (H). For (B right) and (D), Mann–Whitney test was used. For (E), Welch's t-test was used. Data are shown as mean ± SD except for (B), which is shown as mean ± SEM.

tumors, $Kras^{LSL-G12D/WT}$; $p53^{Flox/Flox}$ (KP) mice were intratracheally instilled with $10^8$ pfu Ad5mSPC-Cre (University of Iowa Viral Vector Core facility) viral vector. All mice were bred and maintained in a specific pathogen-free animal facility. Male and female SNL mice and male KP mice were used. All animal studies were approved from the Université Libre de Bruxelles Institutional Animal Care and Use Committee (protocol numbers 01 Gos IBMM, 23 Gos IBMM and 16 Gos CMMI). All experiments were performed in accordance with the approved protocol and other relevant guidelines and regulations.

### Mouse treatments

Mice were treated with A-1331852 (25 mg/kg) formulated in 10% DMSO + 40% PEG300 + 5% Tween-80 + 45% Saline via oral gavage (150–250 μL volume) at indicated timepoints. Control mice were treated with the same volume of vehicle. For chemotherapy, a mixture of carboplatin (10 mg/kg) and paclitaxel (10 mg/kg) was formulated in 50% PEG300 + 50% Saline and injected intraperitoneally (100–200 μL volume) at indicated timepoints. Control mice were treated with the same volume of vehicle. For immunotherapy, 200 μg of anti-PD-1 (clone: 29F.1A12, Bio X Cell) was injected intraperitoneally (125 μL volume) once every two days. Control mice were treated with the same quantity and volume of isotype control antibody (clone 2A3, Bio X Cell).

### Micro-computed tomography and measurement of tumor volume

Mice were first scanned between 10–14 weeks post tumor initiation and randomized for treatment once tumors were measurable. Scanning was performed under isoflurane anesthesia with Bruker Skyscan 1278 at pixel size of 81.78 μm. Reconstruction of scans were done with NRecon software with image-based gating strategy. Measurement of individual tumor size was done blindly in Horos software. Rare tumors with extreme growth rate were excluded from analysis. These outliers were defined as having a growth rate >5-fold in a 2-week time frame prior to any treatment.

### Tissue collection and single-cell processing

Mice were sacrificed with a lethal dose of Pentobarbital. Lung was removed and individual tumors were macro-dissected and cut into fine pieces in RPMI. Tissue digestion was done in GentleMACS tissue dissociator (Miltenyi) supplemented with DNase 1 (0.02 mg/mL, Merck) and collagenase (1 mg/mL, Merck). The digestion reaction was blocked with RPMI supplemented with FBS. Single-cell suspension was obtained after filtering through 70 μm filter. Blood was collected from the heart via cardiac puncture. Bone marrow, blood and spleen single cell suspensions were obtained by lysing with ACK lysis buffer for 10 min at room temperature followed by filtering with 70 μm filter for the bone marrow and spleen samples.

### Histology and immunohistochemistry

At 15-week post-tumor initiation, whole lung or individual tumors were fixed in 10% neutral buffered formalin for 24 h, embedded in paraffin blocks and stored at 4 °C. To classify LUAD and LUSC tumors, H&E sections obtained from FFPE blocks were annotated and assessed by an expert pathologist. Once histological subtype was confirmed, the following slide was then used for MPO staining. FFPE blocks were then deparaffinized, rehydrated, and subjected to antigen retrieval (Citrate, pH 6.1; Agilent) using PT link (Agilent). Sections were then permeabilized in PBS-0.1% Triton X-100 for 10 min followed by blocking with horse serum for 30 min at room temperature. Primary antibody incubation with anti-MPO (1:100, R&D Systems, AF3667) was done in horse serum overnight and rinsed off with PBS-0.05% Tween-20. Labeling secondary antibody and chromogenic reaction with alkaline phosphatase was developed with ImmPRESS®-AP Horse Anti-Goat IgG Polymer Detection Kit (Vector Laboratories, MP-5405) according to the instructions provided with the kit. Finally, sections were counterstained with hematoxylin and scanned with Hamamatsu NanoZoomer S360MD slide scanner with 20x magnification. QuPath's cell classifier was used to quantify MPO positive cells within lesions.

### Flow cytometry

Single-cell suspension of tumor, spleen, and bone marrow was prepared as described and stained with fixable viability dye and Fc-block antibody (1:200, clone 2.4G2) for 25 min at 4 °C in PBS. For identification of bone marrow and splenic progenitors, cells were stained with CD16/32 fluorophore conjugated antibody (1:100) instead of Fc-block. Cell-surface marker staining (See list of antibodies) was done in PBS-BSA (0.5%) for 20 min at 4 °C. For intracellular staining, cells were fixed and permeabilized with BD Cytofix/Cytoperm™ or eBioscience™ Foxp3 Fixation/Permeabilisation kit for 30 min at 4 °C. Staining for Bcl-xL (1:500, Abcam, AB208747) and FoxP3 (1:100, Invitrogen, 15-5773-82) was done for 1 h at 4 °C. Finally, cells were acquired on Beckman Coulter CytoFLEX LX or BD LSRFortessa. Analysis of flow cytometry data was done using FlowJo™ software.

### BrdU labeling and staining

For BrdU labeling, mice were intra-peritoneally injected with 2 mg BrdU (MedChemExpress, HY-15910) dissolved in PBS. Tumors were harvested and processed into single-cell suspension as described. After surface marker staining, fixation was done with eBioscience™ Foxp3 Fixation/Permeabilization kit for 20 min at 4 °C followed by incubation in permeabilization buffer for 30 min at 4 °C. Cells were re-fixed for 15 min followed by incubation with DNAse (300 μg/mL) at 37 °C for 1 h in HBSS. Finally, BrdU staining was done overnight at 4 °C.

**The paper explained**

**Problem**

Immunotherapies are transforming cancer treatment, including for non-small cell lung cancer (NSCLC). However, compared to T cells, the main targets of immunotherapies, innate immune cells including neutrophils have received less attention in the field of cancer. Nevertheless, tumor-associated neutrophils (TANs) constitute a major immune cell fraction in lung adenocarcinoma (LUAD) and lung squamous cell carcinoma (LUSC), the main subtypes of NSCLC. Recent research has highlighted various, even opposing, functions of neutrophils in tumors, making it essential to determine whether and how tumor-supportive neutrophils could be selectively targeted to improve response to conventional therapeutic approaches.

**Results**

By comparing two preclinical mouse models of NSCLC, we revealed molecular and phenotypic similarities but also divergence in TANs from LUAD and from LUSC. Focusing on the latter, we identified that the survival of TANs was augmented compared to that of neutrophils in healthy tissues, and that this depends on a pro-survival protein, Bcl-xL. However, blocking Bcl-xL activity using a highly selective small molecule compound failed to impact tumor development, even when used in combination with anti-PD-1 immunotherapy. Tumors in this model were also relatively resistant to dual carboplatin-paclitaxel chemotherapy. In response to this treatment, we monitored an increase in neutrophils in blood, spleen and tumors, suggesting chemotherapy-induced reactive granulopoiesis. Bcl-xL levels further increased in TANs after chemotherapy, and its blockade synergized with chemotherapy, reducing tumor growth.

**Impact**

We have demonstrated that exploiting a phenotypic disruption of TANs—in this case, the mechanisms controlling their abnormal ageing in cancer—could offer new treatment opportunities considering cells of innate immunity. By integrating results obtained from different mouse models of NSCLC and from several clinically relevant treatments, our data support the notion that the therapeutic impact of targeting tumor-promoting TANs will depend on the type of tumor and the co-administered therapies.

### Single-cell RNA-seq analysis of murine neutrophils

For single-cell RNA-seq (scRNA-seq) analysis comparing TANs to PBNs from SNL mice, published 10X Genomics data in the form of gene expression matrices (stored in Market Exchange Format; MEX format), and TSV files with genes and barcode sequences were downloaded from the Gene Expression Omnibus database (Data ref: Mollaoglu et al, 2018). Data were obtained by high-throughput sequencing of $CD45^+CD11B^+LY6G^+$ neutrophils that were flow sorted and pooled to prepare TANs ($n = 9$ tumors from 2 SNL mice) and PBNs ($n =$ blood from 2 SNL mice). Downloaded data were subject to analyses using a standard Seurat pipeline. Briefly, each dataset was loaded independently into R as a Seurat object using the Read10X and CreateSeuratObject functions. Cells were filtered based on standard quality control (QC) thresholds, including the number of detected genes, total UMI counts, and mitochondrial transcript percentage. After QC, the two datasets representing $n = 1702$ PBNs and 445 TANs were merged into a single Seurat object using the merge function. Following merging, all downstream normalization and scaling steps were performed on the combined object. Gene expression values were normalized using NormalizeData with the default *LogNormalize* method. The expression values were scaled across cells using ScaleData. Normalized or scaled expression values were used for all downstream analyses, including visualization, module scoring, and differential expression. Neutrophil subclass signatures were derived from previously published scRNA-seq data defining murine immune populations and neutrophil heterogeneity (Data ref: Zilionis et al, 2019). Murine N1, N2, N3, N4, N5, and N6 signatures were taken directly from signatures provided in (Zilionis et al, 2019). Seurat's DoHeatmap function was used to generate the expression heatmap comparing PBNs and TANs, and the AddModuleScore function with default settings was used to generate the scores for each neutrophil subset. Split violin plots were generated via the VlnPlot function in Seurat and statistics comparing expression values in PBNs versus TANs were performed in R via Wilcoxon rank-sum testing.

### Human cell line treatments

The FaDu and SK-MES-1 cell lines, provided by Dr. Jawahar Kopparam, the Netherlands Cancer Institute, were tested for mycoplasma contamination (Mycoplasma Detection Kit (Cat. no. 13100-01 SouthernBiotech™) upon arrival and were negative but were not authenticated. 5000 FaDu or SK-MES-1 cells were plated in 96-wells and cultured overnight in MEM with 10% FBS to allow adhesion. On the following day, the medium was replaced with medium containing chemotherapy alone or in combination with A-1331852. After 48 h of incubation, the images were read and confluency of cells was analyzed on Incucyte® SX5.

### Statistics

Sample size was estimated based on the data from previous work (Bodac et al, 2024). Data were exported to GraphPad Prism software for evaluation of statistical significance. Exact *p*-value is reported for each figure. Normality or lognormality of data was assessed using Shapiro-Wilk test. Analysis was done with either Welch's t-test, Mann-Whitney test, Friedman test, one-way or two-way ANOVA depending on the number of groups and normality of data.

### Graphics

Figure 1A and synopsis graphics were created with bioRender.com.

# Data availability

The flow cytometry and micro-computed tomography source data produced in this study are available at Zenodo (https://doi.org/10.5281/zenodo.18257182).

The source data of this paper are collected in the following database record: biostudies:S-SCDT-10_1038-S44321-026-00401-z.

# Peer review information

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

## Acknowledgements

The FaDu and SK-MES-1 cell lines were kindly provided by Dr. Jawahar Kopparam, the Netherlands Cancer Institute. We thank Maëlle Luypaert for maintaining the SNL mouse colony. We thank DIAPath - CMMI for their technical assistance. The CMMI is supported by the European Regional Development Fund and the Walloon Region. This work was supported financially by the Fund for Scientific Research – FNRS under Grants *Ulysse Incentive Grant for Mobility in Scientific Research* [MISU F.6003.22], *Télévie* [TLV 7.4574.21 and 7.6517.23] and *Research Credit* [CDR J.0162.25] (EM), the Belgian Cancer Foundation (project FA/2022/2020 (EM) and Clinical mandate 2023 (MB)), the ULB *Actions de Recherche Concertées* (ARC consolidation) (EM), the Association Jules Bordet (EM), the Swiss National Science Foundation [310030_179324 (EM) and 3200-0-239913 (EM and JYP)], the Fondation Michel Tossizza (EM), and the NIH National Cancer Institute (NCI) under award U24CA213274 (TGO) and R01-CA244841-05 (TGO). AM was supported by *Télévie* and by the Fondation Rose et Jean Hoguet. BP was supported by the Fondation Cancer Hainaut. TGO received support as a Duke Science & Technology Scholar. SG and FA are research director and senior research associate of the Fund for Scientific Research – FNRS, respectively.

## Author contributions

**Abdullah Mayet**: Conceptualization; Data curation; Formal analysis; Validation; Investigation; Visualization; Methodology; Writing—original draft; Writing—review and editing. **Beatrice Parma**: Formal analysis; Investigation. **Déborah Lécuyer**: Formal analysis; Investigation. **Amaury Defruit**: Formal analysis; Investigation. **Sarika Rana**: Investigation. **Anita Bodac**: Methodology. **Pieter Demetter**: Formal analysis; Investigation. **Sébastien Denanglaire**: Investigation. **Abbie S Ireland**: Resources; Data curation; Formal analysis. **Mariana Brandão**: Formal analysis; Funding acquisition. **Thierry Berghmans**: Formal analysis. **Fabienne Andris**: Formal analysis; Funding acquisition. **Stanislas Goriely**: Resources; Formal analysis; Funding acquisition. **Jean Yannis Perentes**: Formal analysis; Funding acquisition. **Trudy G Oliver**: Resources; Formal analysis; Funding acquisition. **Etienne Meylan**: Conceptualization; Formal analysis; Supervision; Funding acquisition; Validation; Visualization; Writing—original draft; Project administration; Writing—review and editing.

Source data underlying figure panels in this paper may have individual authorship assigned. Where available, figure panel/source data authorship is listed in the following database record: biostudies:S-SCDT-10_1038-S44321-026-00401-z.

## Disclosure and competing interests statement

The authors declare no competing interests.

# Expanded View Figures

**Figure EV1.   Variability between TANs in LUSC compared to LUAD.**

(A) (Left) Representative MPO staining and (Right) quantification in LUAD and LUSC lesions of SNL mice ($n = 11$ for LUAD and $n = 6$ for LUSC). Scale bars: 20 μm. (B) Comparison of frequency of CD101$^-$ (left), CD14$^+$ (middle) and PD-L1$^{hi}$ (right) TANs in KP ($n = 8$ tumors) and SNL ($n = 11$–14 tumors). (C) Heatmap depicting expression of individual genes by single-cell RNA-seq (scRNA-seq) of PBNs ($n = 1702$ cells) and TANs ($n = 445$ cells) from SNL mice. Genes are categorized by neutrophil subsets "N1", "N2", "N3", "N4", "N5", and "N6", as defined previously from murine scRNA-seq data (Data ref: Zilionis et al, 2019). (D) Split violin plots depicting expression of indicated neutrophil subset signature in scRNA-seq data of PBNs compared to TANs of SNL mice. Data information: *p*-value was determined using unpaired t-test for (A) and (B left and right), Welch's t-test for (B middle). For (D), Wilcoxon rank-sum test was used (****$p < 3.3E\text{-}0.5$). Data are shown as mean ± SD (A, B). For (D), box and whisker plot overlays depict median and upper and lower quartiles of expression. Each dot represents an individual cell.

▶

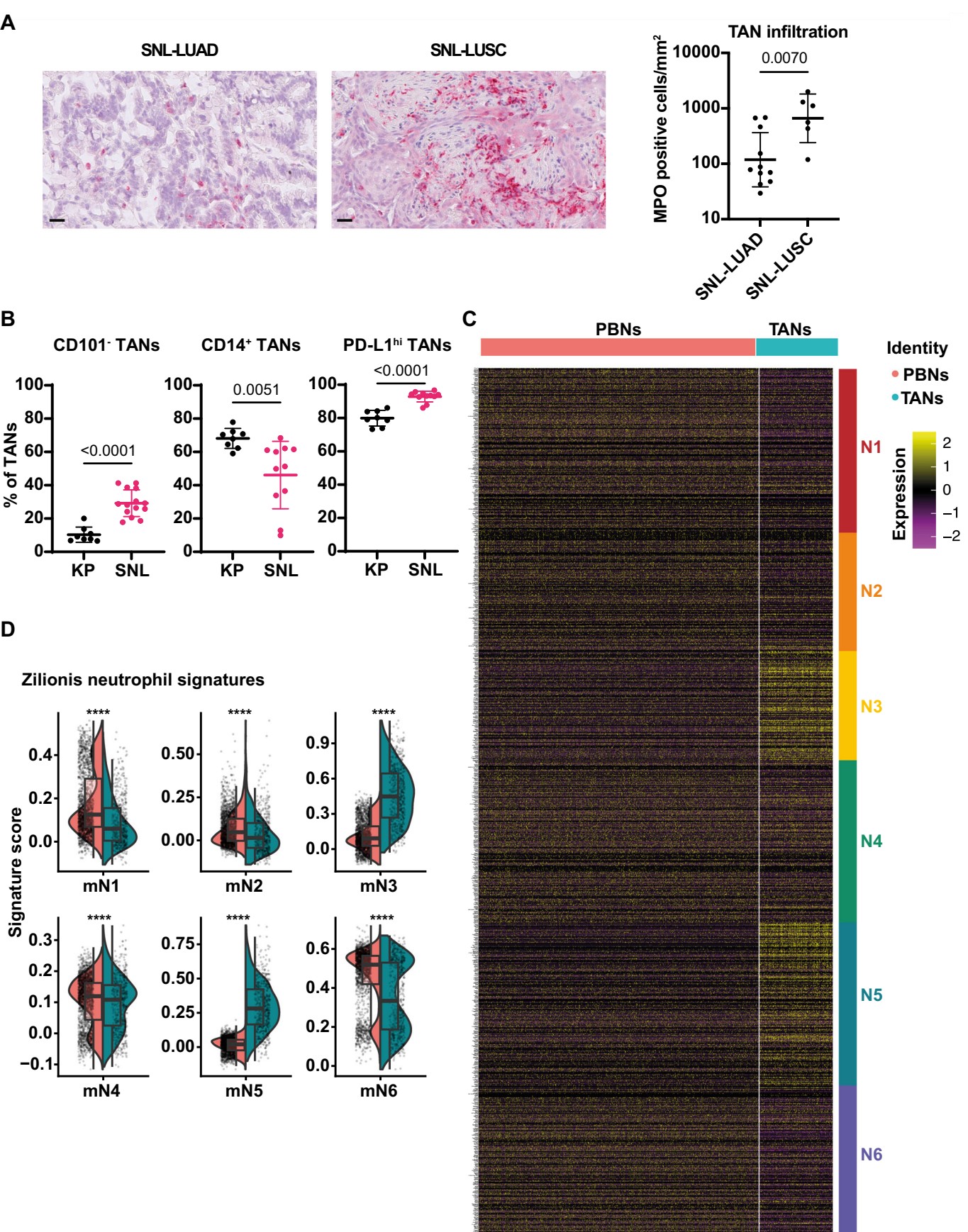

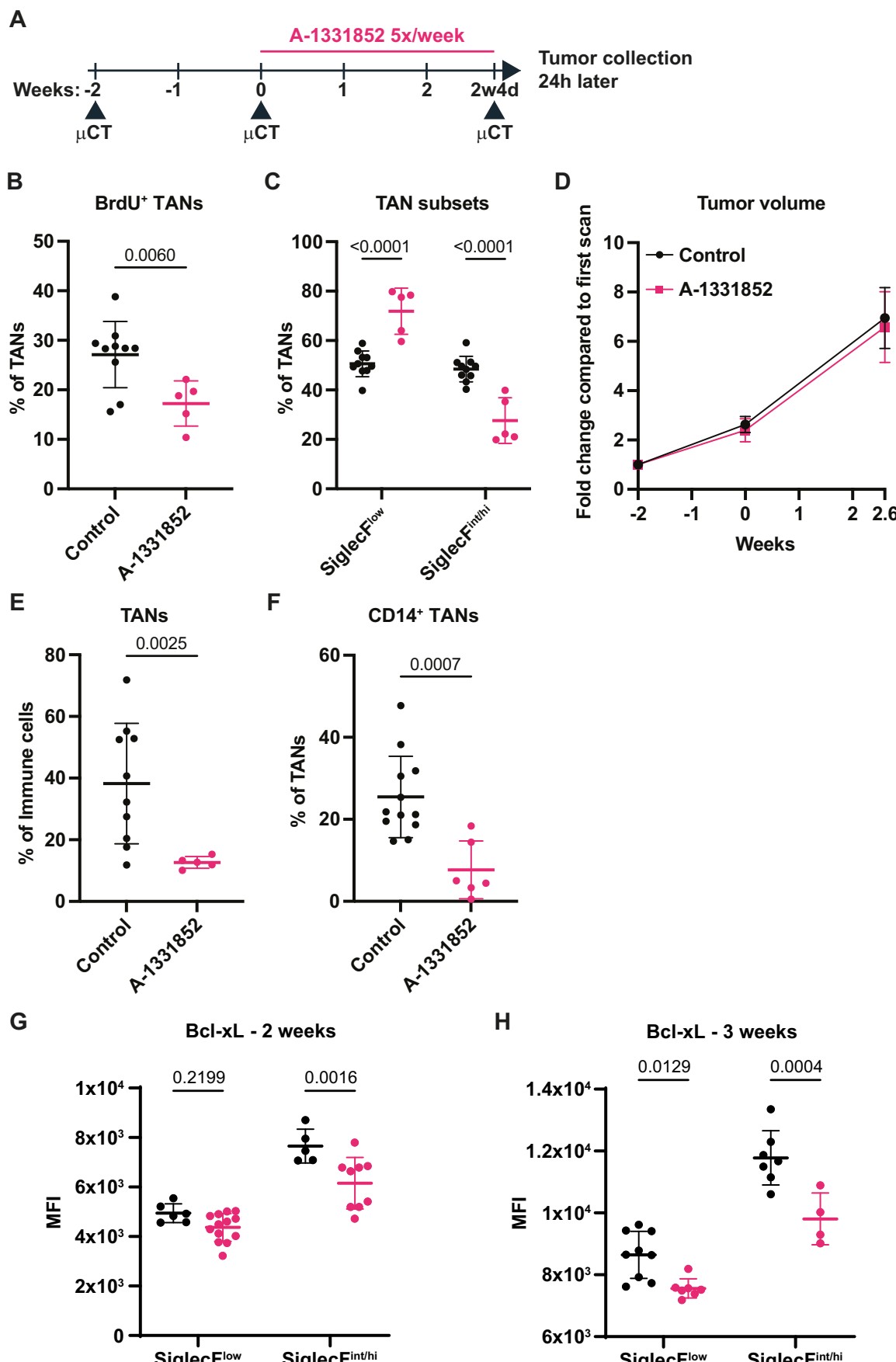

◀ **Figure EV2. Bcl-xL blockade 5x/week does not diminish SNL tumor growth.**

(A) Scheme of in vivo Bcl-xL blockade regimen. (B) Proportions of 6.5-day-old BrdU$^+$ TANs among total TANs ($n = 10$ for control and $n = 5$ for treated group). (C) Proportions of SiglecF TAN subsets among total TANs ($n = 10$ tumors for control and $n = 5$ for treated group). (D) Tumor growth kinetics measured by μCT ($n = 11$ tumors for control group and $n = 6$ tumors for treated group). (E) Proportions of TANs among all immune cells ($n = 10$ tumors for control and $n = 5$ for treated group). (F) Proportions of CD14 TANs among total TANs ($n = 12$ tumors for control and $n = 6$ tumors for treated). (G) MFI of Bcl-xL expression in SiglecF$^{low}$ ($n = 6$ tumors for control and $n = 13$ for treated group) and SiglecF$^{int/hi}$ ($n = 5$ for control and $n = 9$ for treated group) TANs. (H) MFI of Bcl-xL expression in SiglecF$^{low}$ ($n = 9$ tumors for control and $n = 7$ for treated group) and SiglecF$^{int/hi}$ ($n = 7$ for control and $n = 4$ for treated group) TANs. Data information: $p$-value was determined using Welch's t-test for (B), (E), and (F). For (C), (D), (G), and (H), 2-way ANOVA was used. Data are shown as mean ± SD except for (D), which is shown as mean ± SEM.

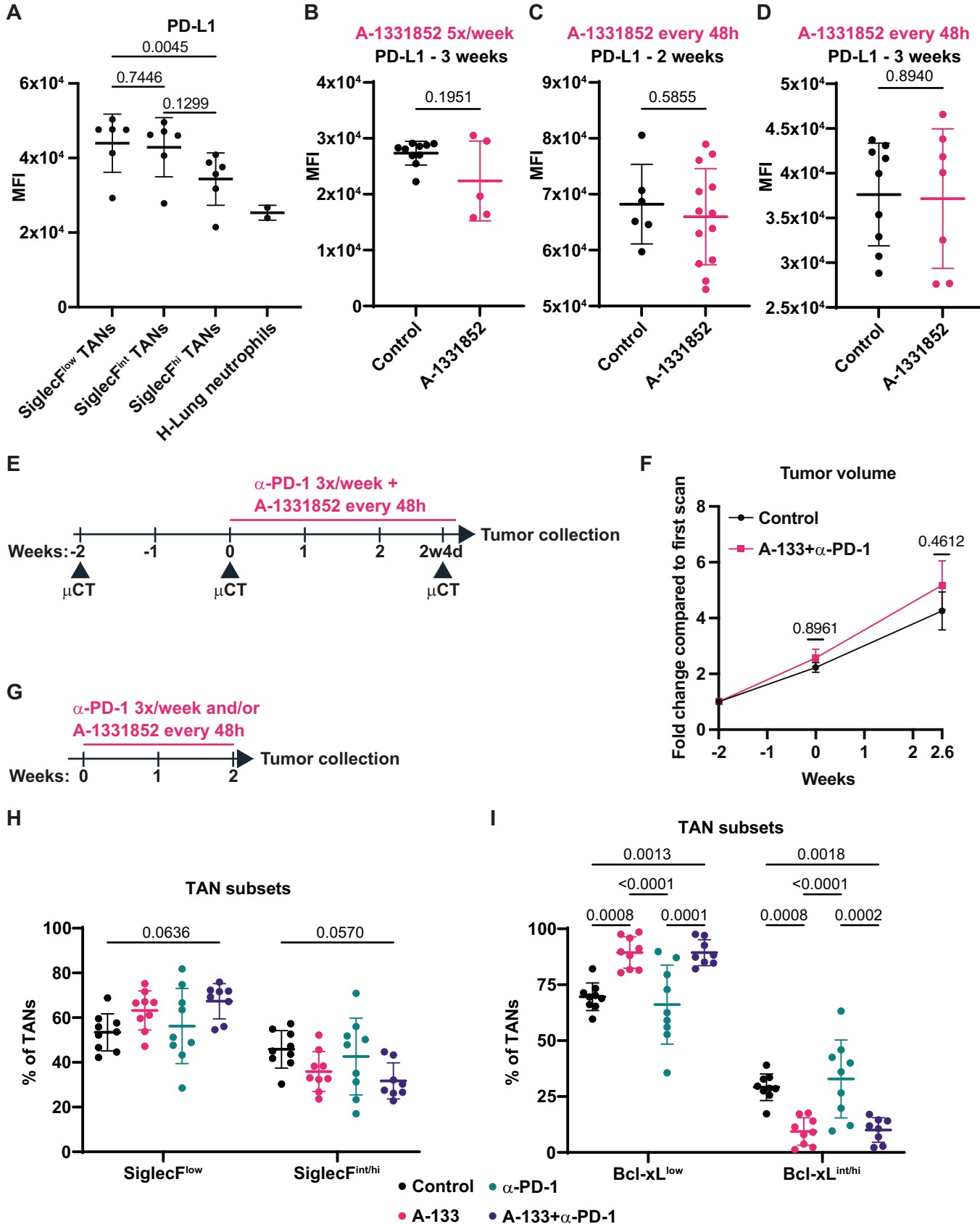

◀ **Figure EV3. Combined Bcl-xL blockade and anti-PD-1 does not diminish tumor growth.**

(A) MFI of PD-L1 expression among TAN subsets ($n = 6$ for matched tumors and $n = 2$ for healthy lung (H-Lung) neutrophils). (B) MFI of PD-L1 expression in TANs upon 5x/week A-1331852 regimen for 3 weeks ($n = 10$ tumors for control and $n = 5$ for treated group). (C) MFI of PD-L1 expression in TANs upon 1x/48 h A-1331852 regimen for 2 weeks ($n = 6$ tumors for control and $n = 13$ for treated group). (D) MFI of PD-L1 expression in TANs upon 1x/48 h A-1331952 regimen for 3 weeks ($n = 9$ tumors for control and $n = 7$ for treated group). (E) Scheme of anti-PD-1 immunotherapy combined with A-1331852. (F) Tumor growth kinetics measured by μCT ($n = 12$ tumors for control group and $n = 8$ tumors for treated group). (G) Scheme of anti-PD-1, A-1331852 or their combination for 2 weeks. (H) Proportion of SiglecF TAN subsets among total TANs ($n = 8$–9 tumors per group). (I) Proportion of Bcl-xL TAN subsets among total TANs ($n = 8$–9 tumors per group). Data information: p-value was determined using Friedman test for (A). For (B), Welch t-test was used and for (C), and (D), unpaired t-test was used. For (F), (H), and (I), 2-way ANOVA was used. Data are shown as mean ± SD except for (F), which is shown as mean ± SEM.

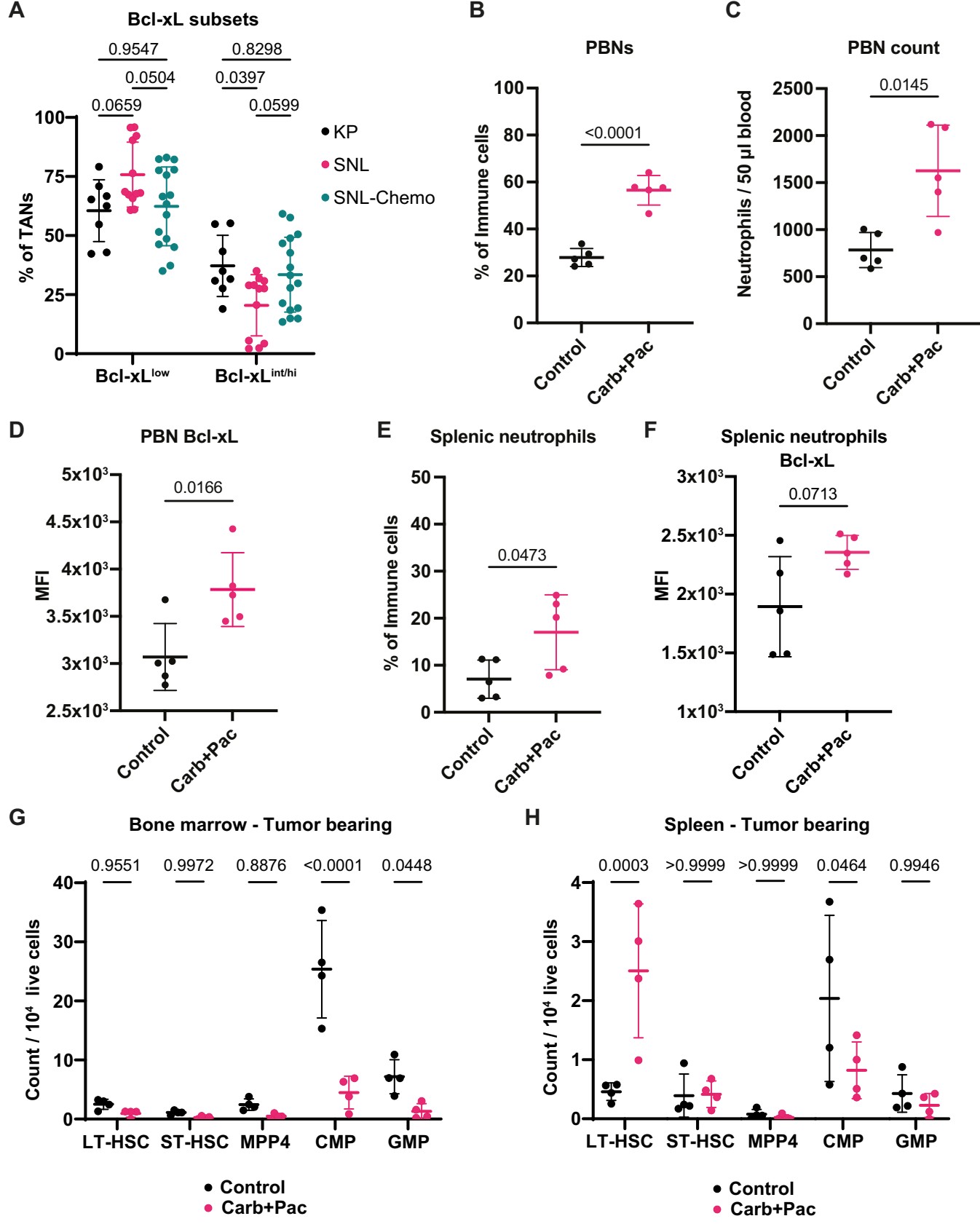

◀  **Figure EV4.  Combination chemotherapy increases circulating neutrophil numbers and their expression of Bcl-xL.**

(A) Proportion of Bcl-xL TAN subsets among total TANs ($n = 8$–16 tumors per group). (B) Proportions of PBNs among immune cells in control group or carboplatin and paclitaxel (Carb+Pac) treated group. (C) Absolute PBN counts per 50 μl of blood. (D) MFI of Bcl-xL expression in PBNs. (E) Splenic neutrophil proportions among immune cells. (F) MFI of Bcl-xL expression in splenic neutrophils. (G, H) Quantities of hematopoietic precursors in the bone marrow (G) and spleen (H) among $10^4$ live cells ($n = 4$ mice per group). Data information: $p$-value was determined using 2-way ANOVA for (A), (G), and (H). For (B), (C), (D), (E), and (F), Welch's t-test was used ($n = 5$ per group). Data are shown as mean ± SD.

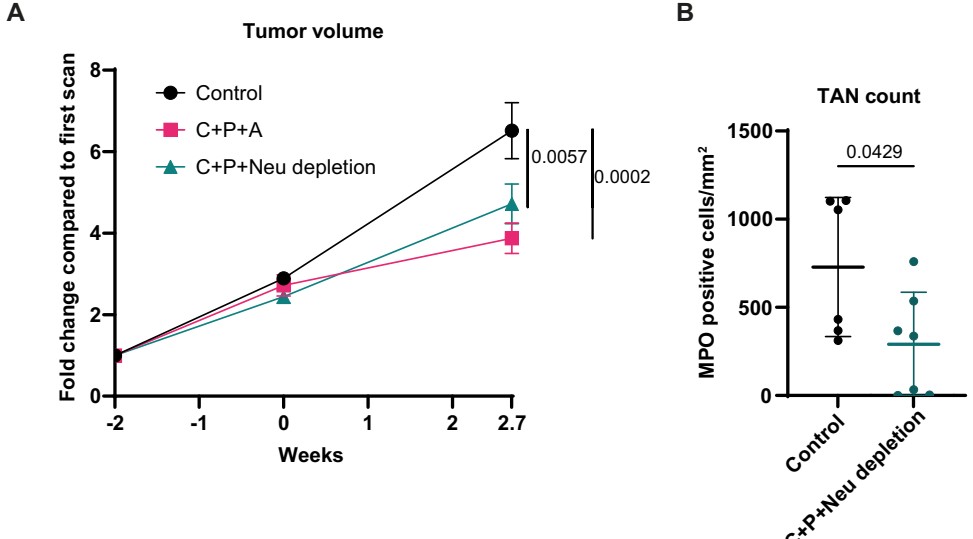

**Figure EV5.** **Partial neutrophil depletion is comparable to A-1331852 in the anti-tumor response to chemotherapy.**

(A) Tumor growth kinetics measured by μCT ($n = 9$–14 tumors per group). (B) IHC Quantification of MPO positive cells per mm² of tumor ($n = 6$ tumors for control and $n = 7$ for neutrophil depleted group). Data information: *p*-value was determined using 2-way ANOVA for (A) and unpaired *t*-test for (B). Data is shown as ±SEM for (A) and ±SD for (B).

