## [Peer Review File · EMBO Molecular Medicine]

Bcl-xL blockade targets neutrophils and synergizes with chemotherapy in lung squamous cell carcinoma

Abdullah Mayet, Beatrice Parma, Déborah Lécuyer, Amaury Defruit, Sarika Rana, Anita Bodac, Pieter Demetter, Sébastien Denanglaire, Abbie Ireland, Mariana Brandão, Thierry Berghmans, Fabienne Andris, Stanislas Goriely, Jean Yannis Perentes, Trudy Oliver, and Etienne Meylan

Corresponding author: Etienne Meylan (Meylan.Etienne@chuv.ch)

Review Timeline:

Submission Date:	17th Sep 25
Editorial Decision:	30th Sep 25
Revision Received:	23rd Jan 26
Editorial Decision:	11th Feb 26
Revision Received:	13th Feb 26
Accepted:	26th Feb 26

Editor: Lise Roth

Transaction Report:

30th Sep 2025

Dear Prof. Meylan,

Thank you for submitting your manuscript to EMBO Molecular Medicine. We have now received feedback from the three reviewers who agreed to evaluate your manuscript. As you will see from the reports below, they acknowledge the potential interest of the findings and the general technical quality of the study, but also make suggestions to improve the manuscript further. If you feel you can satisfactorily address these points experimentally when possible, or by adequate discussion, you may wish to submit a revised version of your manuscript.

Addressing the reviewers' concerns in full will be necessary for further considering the manuscript in our journal, and acceptance of the manuscript will entail a second round of review. EMBO Molecular Medicine encourages a single round of revision only and therefore, acceptance or rejection of the manuscript will depend on the completeness of your responses included in the next, final version of the manuscript. For this reason, and to save you from any frustrations in the end, I would strongly advise against returning an incomplete revision.

We are expecting your revised manuscript within three months, if you anticipate any delay, please contact us.

We require:

Additional information on source data and instruction on how to label the files are available

4) A .docx formatted letter INCLUDING the reviewers' reports and your detailed point-by-point responses to their comments. As part of the EMBO Press transparent editorial process, the point-by-point response is part of the Review Process File (RPF), which will be published alongside your paper.

5) A complete author checklist, which you can download from our author guidelines (<https://www.embopress.org/page/journal/17574684/authorguide#submissionofrevisions>). Please insert information in the checklist that is also reflected in the manuscript. The completed author checklist will also be part of the RPF.

6) All Materials and Methods need to be described in the main text using our 'Structured Methods' format. According to this format, the Methods section includes a Reagents and Tools Table (listing key reagents, experimental models, software and relevant equipment and including their sources and relevant identifiers) followed by a Methods and Protocols section describing the methods, ideally using a step-by-step protocol format. The aim is to facilitate adoption of the methodologies across labs. Please download and fill our Reagents and Tools Table template (.docx), which you can find in our author guidelines: <https://www.embopress.org/page/journal/14693178/authorguide#structuredmethods>.

7) Please note that all corresponding authors are required to supply an ORCID ID for their name upon submission of a revised manuscript.

8) It is mandatory to include a 'Data Availability' section after the Materials and Methods. Before submitting your revision, primary datasets produced in this study need to be deposited in an appropriate public database, and the accession numbers and database listed under 'Data Availability'. Please remember to provide a reviewer password if the datasets are not yet public (see <https://www.embopress.org/page/journal/17574684/authorguide#dataavailability>).

9) For data quantification: please specify the name of the statistical test used to generate error bars and P values, the number (n) of independent experiments (specify technical or biological replicates) underlying each data point and the test used to calculate p-values in each figure legend. The figure legends should contain a basic description of n, P and the test applied. Graphs must include a description of the bars and the error bars (s.d., s.e.m.). Please provide exact p values.

10) Our journal encourages inclusion of *data citations in the reference list* to directly cite datasets that were re-used and obtained from public databases. Data citations in the article text are distinct from normal bibliographical citations and should directly link to the database records from which the data can be accessed. In the main text, data citations are formatted as follows: "Data ref: Smith et al, 2001" or "Data ref: NCBI Sequence Read Archive PRJNA342805, 2017". In the Reference list, data citations must be labeled with "[DATASET]". A data reference must provide the database name, accession number/identifiers and a resolvable link to the landing page from which the data can be accessed at the end of the reference. Further instructions are available at .

11) We replaced Supplementary Information with Expanded View (EV) Figures and Tables that are collapsible/expandable online. EV Figures should be cited as 'Figure EV1, Figure EV2' etc... in the text and their respective legends should be included in the main text after the legends of regular figures.

12) The paper explained: EMBO Molecular Medicine articles are accompanied by a summary of the articles to emphasize the major findings in the paper and their medical implications for the non-specialist reader. Please provide a draft summary of your article highlighting

13) Author contributions: CRediT has replaced the traditional author contributions section because it offers a systematic machine readable author contributions format that allows for more effective research assessment. Please remove the Authors Contributions from the manuscript and use the free text boxes beneath each contributing author's name in our system to add specific details on the author's contribution. More information is available in our guide to authors.

Please also suggest a visual abstract to illustrate your article as a PNG file 550 px wide x 300-600 px high. A cropped portion of this image will serve as thumbnail for the table of content on our webpage.

16) As part of the EMBO Publications transparent editorial process initiative (see our Editorial at <http://embomolmed.embopress.org/content/2/9/329>), EMBO Molecular Medicine will publish online a Review Process File (RPF) to accompany accepted manuscripts.

In the event of acceptance, this file will be published in conjunction with your paper and will include the anonymous referee reports, your point-by-point response and all pertinent correspondence relating to the manuscript. Let us know whether you agree with the publication of the RPF and as here, if you want to remove or not any figures from it prior to publication. Please note that the Authors checklist will be published at the end of the RPF.

I look forward to receiving your revised manuscript.

Yours sincerely,

Lise Roth

***** Reviewer's comments *****

Referee #1 (Remarks for Author):

Extending their previous studies, the authors report shared and distinct characteristics of TANs from two mouse models of NSCLC. Similar to that in KP tumors, increasing proportion of neutrophils from SNL tumors also expressed SiglecF and Bcl-xL, but unlike KP tumors, SNL tumors remained unresponsive to anti-Bcl-xL therapy. SNL tumors were relatively resistant to combination chemotherapy, or to an anti-PD1 immunotherapy with A1331852, whereas combining carboplatin/paclitaxel with A-1331852 reduced SNL tumors. Chemotherapy enhanced Bcl-xL expression in neutrophils and TANs. These findings would support a link between chemoresistance and neutrophil function, and raise the intriguing possibility of targeting TANs as part of a combination therapy to sensitize SNL tumors to chemotherapy. The following issues may, however, require further attention.

Specific Comments.

1. Arguably, TANs may have distinct phenotypic/functional characteristics in KP and SNL tumors beyond expressing SiglecF and Bcl-xL, for anti-Bcl-xL therapy was inefficient in SNL tumor. Including some additional information on TAN phenotype from KP and SNL tumors would strengthen the manuscript.
2. Bcl-xL blockade in SNL reduced the abundance of CD14+ TANs. Did the authors study other phenotype markers? It is unclear how CD14+ TANs could serve as a biomarker for tumor response to neutrophil-targeted therapies.
3. Carboplatin+paclitaxel+A-1331852 therapy markedly reduced Bcl-xL expression in TANs regardless of the level of SiglecF expression, thereby likely affecting the lifespan of all TANs. How could then the chemotherapy sensitizing effect of Bcl-xL blockade be attributed solely to facilitating removal of aged SiglecFhigh/Bcl-xLhigh TANs? Along this line, the combination chemotherapy also reduced B cell number. What happened to T cells?
4. What was the expression level of Bcl-xL in SNL tumor cells in response to chemotherapy? How could one exclude a direct effect of A1331852 on tumor cells?
5. It is uncertain whether combination chemotherapy triggered reactive granulopoiesis and/or reprogramming TANs in situ.

Minor Points.

1. Lines 290 and 293. "intubated" appears to refer to "instillation".
2. The volume for each drug administered and the appropriate controls should be indicated.
3. Line 317. Please rephrase "terminally anesthetized".

Referee #2 (Remarks for Author):

For my part, the study is impeccable, I have no questions for the authors, and I recommend its publication. However, since there are many technical aspects and I cannot make a judgment on the mouse experiments, I recommend that it also be reviewed by an expert.

Referee #3 (Remarks for Author):

In this manuscript, Mayet et al. investigate the role of tumor-associated neutrophils (TANs) in lung squamous cell carcinoma (LUSC). Building on their previous work showing that aged TANs in lung adenocarcinoma (LUAD) can be selectively eliminated by Bcl-xL inhibition, thereby reducing tumor growth, the authors now explore whether the same approach applies to LUSC. Using the SNL mouse model (Rosa26^{LSL-Sox2-IRES-GFP}; Nkx2-1^{fl/fl}; Lkb1^{fl/fl}), which recapitulates LUAD-to-LUSC transition, they find that although Bcl-xL^{high} TANs accumulate in LUSC, Bcl-xL inhibition alone does not slow tumor progression. However, when combined with standard chemotherapy (carboplatin + paclitaxel), Bcl-xL blockade sensitizes tumors to treatment, leading to reduced tumor growth. This suggests that the therapeutic impact of targeting neutrophils depends on tumor subtype and treatment context.

Overall, the manuscript is well designed and clearly written, and it would be a strong fit for EMBO Molecular Medicine once the authors address the following minor comments. While some of these points may not require additional experiments, they should at least be acknowledged and discussed.

- 1) Since SNL mice can present both LUAD-like and LUSC lesions, it would be highly informative to assess SiglecF and Bcl-xL expression in TANs from pre-lesions versus transformed lesions if technically possible. This could be achieved by combining flow cytometry/IF/IHC analysis with histological annotation to directly link TAN phenotype to tumor histotype.
- 2) While KP mice provide a clean LUAD model, it would be valuable to test whether Bcl-xL inhibition impacts LUAD lesions differently than LUSC lesions within the same SNL mouse, if technically feasible (using cytometry/IF analysis with histological annotation to directly link TAN phenotype to tumor histotype). This could strengthen the mechanistic link between neutrophils and histological fate.
- 3) The BrdU data nicely demonstrate increased TAN lifespan, but it would be useful to repeat this analysis under the reduced-dose Bcl-xL inhibition protocol (Fig. 2A).
- 4) The contribution of Bcl-xL inhibition to tumor control might extend beyond TANs as it could impact cancer cells as well? Testing Bcl-xL inhibition in cancer cells alone (in vitro) or in Ly6G-depleted mice {plus minus} A-1331852 would help disentangle TAN-dependent from tumor cell-intrinsic effects.
- 5) It is unclear in Fig. 2 why there are differences in neutrophil populations at week 2 vs week 3, it should be discussed. Also the authors should indicate Bcl-xL expression in different TANs subtypes (based on siglecF expression).
- 6) TAN aging markers (BrdU, SiglecF, Bcl-xL levels) should be assessed under anti-PD1, A-1331852, or combination treatments. This might explain why certain regimens (e.g. PD1 + Bcl-xL inhibition) failed to slow tumor growth. Additionally, do these treatment affect differently LUAD and LUSC lesions in SNL mice? (if technically possible)
- 7) Since multi-color flow cytometry data are available, the authors could examine immune cell recruitment changes under PD1 blockade. This would help clarify why PD1 treatment fails in SNL tumors.
- 8) Complementing FACS data with immunofluorescence of neutrophil distribution could reveal whether chemotherapy alters TAN localization within tumors in the different context.
- 9) Related to Fig. 3F, it would be useful to stratify Bcl-xL expression by SiglecF^{low} vs SiglecF^{hi} TANs after chemotherapy, to confirm whether aged TANs are preferentially upregulated.
- 10) As Ly6G is a common neutrophil marker, are all TAN subsets equally Ly6G⁺ in the SNL model (based on SiglecF and Bcl-xL expression)? Depleting Ly6G⁺ cells could clarify whether different TAN subsets contribute differently to tumor progression.
- 11) The data suggest that chemotherapy induces more "aggressive" Bcl-xL^{high} TANs, yet tumors only become sensitive when these are removed. This raises the following questions:
 - Does chemotherapy push TANs into a "super-aged" state, for example do TANs live longer post-chemotherapy (BrdU labeling)?
 - Are Bcl-xL levels in chemo-treated SNL TANs comparable to those in KP LUAD TANs (where Bcl-xL inhibition works as monotherapy)?
- 12) In their previous study, the authors show that the use of the anti-neutropenic agent G-CSF potentiates the anti-tumor effect of Bcl-xL inhibition. What would be the effect of G-CSF in this study considering that patients will likely get G-CSF in addition to chemotherapy
- 13) In general, it is unclear why TANs are sometimes classified as SiglecF low vs int vs high and sometimes low vs int/high
- 14) The manuscript does not explore how the N1-N6 neutrophil states, described in LUAD, translate into the LUSC context. It is not a problem, but it would be interesting to clarify whether distinct TAN states are similarly conserved in SNL tumors if the authors have the data.

Referee #1 (Remarks for Author):

Extending their previous studies, the authors report shared and distinct characteristics of TANs from two mouse models of NSCLC. Similar to that in KP tumors, increasing proportion of neutrophils from SNL tumors also expressed SiglecF and Bcl-xL, but unlike KP tumors, SNL tumors remained unresponsive to anti-Bcl-xL therapy. SNL tumors were relatively resistant to combination chemotherapy, or to an anti-PD1 immunotherapy with A1331852, whereas combining carboplatin/paclitaxel with A-1331852 reduced SNL tumors. Chemotherapy enhanced Bcl-xL expression in neutrophils and TANs. These findings would support a link between chemoresistance and neutrophil function, and raise the intriguing possibility of targeting TANs as part of a combination therapy to sensitize SNL tumors to chemotherapy. The following issues may, however, require further attention.

We thank Reviewer for the critical assessment of our research manuscript. To address the specific comments, we have performed several new experiments and analyses.

Specific Comments.

1. Arguably, TANs may have distinct phenotypic/functional characteristics in KP and SNL tumors beyond expressing SiglecF and Bcl-xL, for anti-Bcl-xL therapy was inefficient in SNL tumor. Including some additional information on TAN phenotype from KP and SNL tumors would strengthen the manuscript.

We agree with Reviewer that TANs in the two models might phenotypically differ. First, to address this already at the molecular level, we did a bioinformatics analysis of the KP mN1-mN6 neutrophil signature genes on the SNL model, and identified similarities between the models, including enrichment of mN3 and mN5 in SNL TANs compared to PBNs, and enrichment of mN1 and mN2 in SNL PBNs compared to TANs. However, it was harder to conclude for mN4 and for mN6 showing a bi-modal distribution. See below and also the response to Reviewer 3, Q14. This new data is now new Figure EV1C-D. We added new text, p.7.

To provide additional information about TAN phenotypes from KP and SNL tumors beyond SiglecF and Bcl-xL, we monitored, from all TANs, the expression of CD101, a marker of

mature neutrophils, CD14, an activation marker and LPS co-receptor, and PD-L1. Compared to only 10.3% of KP TANs, 29.2% of SNL TANs were CD101 negative, suggesting a lower proportion of mature neutrophils within SNL tumors. This is in line with the expression of CD14, measured in 68.1% KP TANs compared to only 46.1% SNL TANs. While PD-L1^{hi} TANs are predominant in KP tumors (79.9%), they are even more abundant in SNL TANs (92.9%). Each marker is therefore very significantly differently expressed in TANs from KP vs SNL mice, highlighting phenotypic differences in these cells, at least when considered globally. We added this new data below and as new Figure EV1B. Text was added on p.6-7.

2. Bcl-xL blockade in SLN reduced the abundance of CD14+ TANs. Did the authors study other phenotype markers? It is unclear how CD14+ TANs could serve as a biomarker for tumor response to neutrophil-targeted therapies.

To respond to this question, we have measured PD-L1 protein expression by flow cytometry, in response to Bcl-xL blockade 1x/48h - two weeks of treatment, Bcl-xL blockade 5x/week - three weeks of treatment, and Bcl-xL blockade 1x/48h - three weeks of treatment. There was no significant change in this phenotype marker in any of the studied condition. We have added these data as new Fig. EV3B-D, with new text (p.9).

We have also toned down our sentence on CD14+ TANs potentially serving as biomarker of tumor response to neutrophil-targeted therapies, replacing it by “For a successful tumor response to neutrophil-mediated therapies, it may therefore be crucial to identify ways of maintaining or increasing the abundance of one or more specific TAN subsets, for example those expressing CD14 and certain ISGs.” (p.12).

3. Carboplatin+paclitaxel+A-1331852 therapy markedly reduced Bcl-xL expression in TANs regardless of the level of SiglecF expression, thereby likely affecting the lifespan of all TANs. How could then the chemotherapy sensitizing effect of Bcl-xL blockade be attributed solely to facilitating removal of aged SiglecF^{high}/Bcl-xL^{high} TANs?

Reviewer raised the possibility that the lifespan of all TANs could be affected by the combination (C+P+A) treatment. Indeed, we have now quantified that the total number of TANs per mg of tumor tissue is significantly reduced in response to C+P+A treatment. This data is now displayed as new Fig. 4E and shown here below:

We have added a corresponding text (p.11).

Under chemotherapy treatment, there is higher Bcl-xL expression in circulating neutrophils and in TANs, possibly due to reactive granulopoiesis. In this case, we think Bcl-xL blockade-mediated chemotherapy sensitization might not be solely due to removal of aged SiglecF^{high} TANs, but to a more global removal of TANs expressing higher levels of Bcl-xL, regardless of SiglecF expression. Indeed, we observe reduced MFI for Bcl-xL in all SiglecF subsets upon C+P+A (see Fig. 4G). We therefore reasoned that the chemosensitizing effect of A-133 could be recapitulated by neutrophil depletion. To directly test this possibility, we depleted neutrophils using our established methodology (dual antibody approach with anti-Ly6G + anti-rat; PMID: 32488020), concomitantly with C+P. This reduced tumor growth comparably to C+P+A. This result reinforces that chemotherapy treatment expands the pool of Bcl-xL^{high} neutrophils that impair the efficacy of chemotherapy. Chemotherapy sensitization can be similarly achieved by Bcl-xL blockade or by reduction of total TANs. This new data is presented below and as new Fig. EV5A. The confirmation of reduced TANs in LUSC lesions upon anti-Ly6G + anti-rat treatment was obtained by immunofluorescence using anti-MPO staining; it is shown as new Fig. EV5B. Related text has been added (p.11).

Along this line, the combination chemotherapy also reduced B cell number. What has happened to T cells?

The data on T cells, shown in Fig. 3D, showed no difference in each of NKT, CD4, CD8 and Tregs. For clarification, we provide here below for Reviewer the part of the graph specifically showing the T cell populations.

4. What was the expression level of Bcl-xL in SNL tumor cells in response to chemotherapy? How could one exclude a direct effect of A1331852 on tumor cells?

To address the first question, we are now providing the MFI of Bcl-xL in tumor cells (EPCAM⁺GFP⁺), after three cycles of chemotherapy compared to controls. This shows that, in contrast to TANs (see Fig. 3F, and New Fig. 3H), there is no change of Bcl-xL expression in tumor cells after this treatment. We add this new data here below and as new Fig. 3G.

For the 2nd question, we do not currently exclude an effect of A-1331852 on tumor cells. We provide *in vitro* data showing that the efficacy of increasing doses of carboplatin or paclitaxel is each increased in two human squamous tumor cell lines upon co-incubation with 10 nM A-133. This data suggests that, *in vivo*, A-133 combined with chemotherapy may exert its effects partially directly on tumor epithelial cells. We added this data here below and as new Appendix Figure S4. Accompanying text was added (p.11).

5. It is uncertain whether combination chemotherapy triggered reactive granulopoiesis and/or reprogramming TANs in situ.

Although we cannot exclude that there is TAN reprogramming, our data of enhanced neutrophils in circulation, spleen and tumors suggest there is a systemic response to the combination chemotherapy that is initiated remotely from the tumors, i.e. in the BM or the spleen. To further examine this possibility, we measured the frequencies of hematopoietic stem or progenitor cells, specifically LT-HSC, ST-HSC, MPP4 (lymphoid-primed progenitor), CMP and GMP after three cycles of Carb+Pac compared to control treated animals. We reasoned that a chemotherapy-mediated reactive granulopoiesis would be associated with quantifiable changes in one or some of these cell populations. In the BM, there was a significant reduction of CMPs and GMPs. This could be due to these cellular pools being used and consumed in response to chemotherapy treatment, enabling the increased generation of neutrophils, or being particularly sensitive to chemotherapy. In the spleen, CMPs were also reduced. Specifically in the spleen, we monitored a very significant increased LT-HSCs after chemotherapy compared to untreated mice. Increased splenic LT-HSCs may contribute to support or increase the rate of new neutrophil generation at this extramedullary site, under the pressure imposed by chemotherapy. In a future research project, we plan to perform additional follow-up experiments with different durations of treatment and a closer monitoring of the state and proliferation of additional neutrophil progenitor cells.

We add this new data here below and as new Figure EV4G-H. We added new text (p.10). We provide the gating strategy as new Appendix Figure S3.

Minor Points.

1. Lines 290 and 293. "intubated" appears to refer to "instillation".
We have replaced "intubated" by "instilled" at both places (p.16).

2. The volume for each drug administered and the appropriate controls should be indicated.
These were added (p.16).

3. Line 317. Please rephrase "terminally anesthetized".

We have replaced "Mice were terminally anaesthetized with Pentobarbital." by "Mice were sacrificed with a lethal dose of Pentobarbital." (p.17).

Referee #2 (Remarks for Author):

For my part, the study is impeccable, I have no questions for the authors, and I recommend its publication. However, since there are many technical aspects and I cannot make a judgment on the mouse experiments, I recommend that it also be reviewed by an expert.

We thank Reviewer 2 for the positive evaluation of our manuscript.

Referee #3 (Remarks for Author):

In this manuscript, Mayet et al. investigate the role of tumor-associated neutrophils (TANs) in lung squamous cell carcinoma (LUSC). Building on their previous work showing that aged TANs in lung adenocarcinoma (LUAD) can be selectively eliminated by Bcl-xL inhibition, thereby reducing tumor growth, the authors now explore whether the same approach applies to LUSC. Using the SNL mouse model (Rosa26^{LSL}-Sox2-IRES-GFP; Nkx2-1^{fl/fl}; Lkb1^{fl/fl}), which recapitulates LUAD-to-LUSC transition, they find that although Bcl-xL^{high} TANs accumulate in LUSC, Bcl-xL inhibition alone does not slow tumor progression. However, when combined with standard chemotherapy (carboplatin + paclitaxel), Bcl-xL blockade sensitizes tumors to treatment, leading to reduced tumor growth. This suggests that the therapeutic impact of targeting neutrophils depends on tumor subtype and treatment context.

Overall, the manuscript is well designed and clearly written, and it would be a strong fit for EMBO Molecular Medicine once the authors address the following minor comments. While some of these points may not require additional experiments, they should at least be acknowledged and discussed.

We thank Reviewer for the careful assessment of our manuscript and the constructive criticisms. In response to the following minor comments, we have performed several additional *in vitro* and *in vivo* experiments. We have unfortunately encountered a technical problem from our stocks of BrdU, which prevented us from revealing BrdU+ TANs during the revision process. Instead, the corresponding points are acknowledged and discussed.

1) Since SNL mice can present both LUAD-like and LUSC lesions, it would be highly informative to assess SiglecF and Bcl-xL expression in TANs from pre-lesions versus transformed lesions if technically possible. This could be achieved by combining flow cytometry/IF/IHC analysis with histological annotation to directly link TAN phenotype to tumor histotype.

We thank Reviewer for this comment. At time of sacrifice, it should be noted that most lesions from the SNL model are LUSC and not LUAD, making the comparison difficult. Nevertheless, we first attempted to use fixed SNL lung tissue sections with anti-MPO (neutrophils), anti-Bcl-xL and anti-SiglecF antibodies for multiplex-immunofluorescence staining. However, the quality of staining with anti-Bcl-xL and anti-SiglecF was not good enough and the results not quantifiable.

For flow cytometry, we usually use the entire tumor material for staining, making it hard to determine tumor histotype. However, from a small experiment, we have been able to separate tumors into two parts, for H&E histological reading and for flow cytometry. From this specific case, we were able to identify and compare three LUAD to seven LUSC tumors from SNL mice. Using an anti-Bcl-xL antibody did not reveal any significant difference in expression according to tumor histotype. To reach a definitive conclusion, more tumors should be analyzed in follow-up studies. We add the data here below specifically for Reviewer:

2) While KP mice provide a clean LUAD model, it would be valuable to test whether Bcl-xL inhibition impacts LUAD lesions differently than LUSC lesions within the same SNL mouse, if technically feasible (using cytometry/IF analysis with histological annotation to directly link TAN phenotype to tumor histotype). This could strengthen the mechanistic link between neutrophils and histological fate.

We agree with Reviewer that having the possibility, within the same model, to dissect responses of LUAD vs LUSC to the same drug is very interesting. Unfortunately, this appears to be too challenging here because most tumors at advanced stage are LUSC in the SNL model. We think that other models with a more equal distribution of NSCLC histology at advanced stage will be more suitable to directly address this question.

3) The BrdU data nicely demonstrate increased TAN lifespan, but it would be useful to repeat this analysis under the reduced-dose Bcl-xL inhibition protocol (Fig. 2A).

As indicated above, we have tried this experiment for the revision, but had a technical problem with BrdU, which prevented us from detecting any positive signal. Because SiglecF is a good biomarker of TAN ageing (see Fig. 1E), SiglecF^{int/hi} TANs being similarly diminished as BrdU+ TANs upon A-133 5x/week (see Fig. EV2B-C), we anticipate observing a similar reduction of BrdU signal compared to SiglecF^{int/hi} (see Fig. 2B), upon A-133 every 48h.

4) The contribution of Bcl-xL inhibition to tumor control might extend beyond TANs as it could impact cancer cells as well? Testing Bcl-xL inhibition in cancer cells alone (in vitro) or in Ly6G-depleted mice {plus minus} A-1331852 would help disentangle TAN-dependent from tumor cell-intrinsic effects.

We agree with Reviewer that cancer cells might be sensitive to Bcl-xL blockade. To address this point, we performed an *in vitro* experiment. We used two human squamous tumor cell lines, FaDu and SK-MES-1, and applied A-1331852 at a fixed dose of 10 nM, together with increasing doses of each of the chemotherapy used *in vivo*, carboplatin or paclitaxel. We used % of cell confluency as a measure of the cytotoxic effects of the drugs. For each cell line, A-133, while not cytotoxic at the lowest doses of chemotherapy used, increased the cellular cytotoxicity of carboplatin and paclitaxel. We conclude that the anti-tumor effects of blocking Bcl-xL in addition to chemotherapy might arise from the targeting of multiple cell

types, including cancer cells and TANs. We have added this new data (and associated methods) here below and as new Appendix Figure S4. We have updated the text accordingly (p.11).

5) It is unclear in Fig. 2 why there are differences in neutrophils populations at week 2 vs week 3, it should be discussed. Also the authors should indicate Bcl-xL expression in different TANs subtypes (based on siglecF expression).

As noted by Reviewer, at week 2 of A-133 treatment, we observed a selective reduction in SiglecF(int/high) TANs (Fig. 2B), but by week 3 this was not seen anymore (Fig. 2H). Possibly, under longer selection pressure due to treatment, Bcl-xL induction and dependency begin earlier in neutrophils, prior to their increase of SiglecF observed in tumors. Thus, prolonged Bcl-xL blockade might lead to a separation between Bcl-xL induction/dependency and SiglecF expression levels.

This is in line with our response to the second part of this comment: we now indicate Bcl-xL expression in SiglecF(low) vs. SiglecF(int/high) subsets. At 2 weeks of A-133 treatment, we observe a diminished Bcl-xL expression selectively in the SiglecF(int/high) subset, whereas at 3 weeks the decrease is significant in both subsets. We provide this data here below for Reviewer and as New Data Fig. EV2G-H. We have added a corresponding sentence in the text (p.8).

A-1331852 every 48h

6) TAN aging markers (BrdU, SiglecF, Bcl-xL levels) should be assessed under anti-PD1, A-1331852, or combination treatments. This might explain why certain regimens (e.g. PD1 + Bcl-xL inhibition) failed to slow tumor growth. Additionally, do these treatment affect differently LUAD and LUSC lesions in SNL mice? (if technically possible)

We have now assessed Bcl-xL and SiglecF in TANs after a two-week treatment with either A-1331852, anti-PD-1 or both. Whenever A-133 was used, there was a trend toward increased/decreased proportions of SiglecF^{low}/SiglecF^{int/hi} TANs, and a very significant increase/decrease of Bcl-xL^{low}/Bcl-xL^{int/hi} TANs. Anti-PD-1 alone did not alter these proportions and did not impact the changes due to A-133. The data are shown below and as New Fig. EV3H-I. New text is provided (p.9).

Concerning the last part of this question, we do not know, as most lesions at end point are LUSC, rendering such an experiment technically too challenging.

7) Since multi-color flow cytometry data are available, the authors could examine immune cell recruitment changes under PD1 blockade. This would help clarify why PD1 treatment fails in SNL tumors.

We followed Reviewer's advice and performed a multi-color flow cytometry analysis reporting the proportions of the main tumor-infiltrating immune cells after two weeks of control, A-133, anti-PD-1, or A-133 + anti-PD-1 treatment. Total TANs did not change in response to A-133 alone, confirming our previous data shown in Fig. 2C. Among all immune cells examined, anti-PD-1 significantly diminished the proportion of total TANs, while the other immune cells were not significantly affected. Although the lack of tumor-infiltrating immune cell variations may help explain the lack of response of SNL tumors to anti-PD-1, additional tumor, stromal and immune cell phenotyping should be performed in subsequent work to better understand the non-efficacy of this immunotherapy. We have added this new data below and as new Appendix Fig. S2. New text is provided (p.9).

8) Complementing FACS data with immunofluorescence of neutrophil distribution could reveal whether chemotherapy alters TAN localization within tumors in the different context.

To address this comment, we have stained SNL tumor sections with anti-MPO antibodies to reveal TANs, comparing vehicle-treated mice to mice treated with carboplatin + paclitaxel according to our protocol shown in Fig. 3A (three cycles, 1x/week). Upon examination of several tumors from each condition, we did not notice major changes in TAN intratumoral localization. We provide below, for reviewer, two illustrative examples, each one from one of the two conditions:

Localisation of TANs in tumors from control vs. chemotherapy treated mice.

Anti-MPO staining (TANs) with heamatoxylin counterstain of two representative SNL tumors from vehicle treated mice (left) or from mice treated with three cycles of weekly carboplatin and paclitaxel (right). Scale bars: 100 μm.

9) Related to Fig. 3F, it would be useful to stratify Bcl-xL expression by SiglecF^{low} vs SiglecF^{hi} TANs after chemotherapy, to confirm whether aged TANs are preferentially upregulated.

We have now stratified Bcl-xL expression by SiglecF^{low} vs SiglecF^{int/high} TANs after chemotherapy. This shows that there is a selective increase of Bcl-xL expression in SiglecF^{low} but not in SiglecF^{int/high} TANs. We interpret this as chemotherapy-mediated reactive granulopoiesis possibly generating neutrophils that increase Bcl-xL expression prior to SiglecF, in contrast to untreated conditions. This sustains our claim in the Discussion that TANs are phenotypically different in untreated vs chemotherapy treated conditions. We provide this data here below and as new Figure 3H. We have updated the text on p. 9 and p. 13.

10) As Ly6G is a common neutrophil marker, are all TAN subsets equally Ly6G⁺ in the SNL model (based on SiglecF and Bcl-xL expression)? Depleting Ly6G⁺ cells could clarify whether different TAN subsets contribute differently to tumor progression.

To address this question, we have quantified by flow cytometry Ly6G expression in TANs based on SiglecF and Bcl-xL. By flow cytometry, we detect a trend toward higher membrane Ly6G protein expression in SiglecF^{int/high} compared to SiglecF^{low}, and a close to significant increase of Ly6G in Bcl-xL^{int/high} compared to Bcl-xL^{low} TANs. However, it should be noted that Ly6G expression is elevated in all TANs. To illustrate this, we have replaced the former SiglecF/FSC-A pseudocolor plot by a new, SiglecF/LY6G one, in Fig. 1B (left).

The LY6G MFI data is provided here below for Reviewer:

To address the 2nd part of this comment, we have depleted neutrophils using our protocol [anti-Ly6G + anti-rat] (PMID: 32488020), to monitor how this would compare to the chemotherapy-sensitizing effect of A-1331852. Partial depletion of neutrophils (verified in tumors with anti-MPO staining) in combination to carboplatin and paclitaxel reduced tumor growth comparably to C+P+A. Given the elevated Ly6G expression across TANs despite a trend toward higher levels in Bcl-xL^{int/high} cells, we favor the possibility that, under chemotherapy treatment, there is an expanded pool of TANs expressing Bcl-xL and counteracting the effect of therapy; targeting them with A-1331852 or with neutrophil depleting antibodies enhances the efficacy of chemotherapy. This new data is presented below, and as new Fig. EV5A-B. Related text has been added, p.11.

11) The data suggest that chemotherapy induces more "aggressive" Bcl-xL^{high} TANs, yet tumors only become sensitive when these are removed. This raises the following questions:
 - Does chemotherapy push TANs into a "super-aged" state, for example do TANs live longer post-chemotherapy (BrdU labeling)?

We have unfortunately not been able to directly measure if TANs live longer post-chemotherapy yet. Based on the changes of Bcl-xL expression already in blood neutrophils upon chemotherapy treatment (see Fig. EV4D) and the associated systemic response, we favor the possibility of an earlier Bcl-xL increase in neutrophils, and a stronger dependency on it. We do not, however, rule out a super-aged TAN state upon chemotherapy, as the further increase of Bcl-xL might indeed render them resistant to apoptosis for prolonged durations.

- Are Bcl-xL levels in chemo-treated SNL TANs comparable to those in KP LUAD TANs (where Bcl-xL inhibition works as monotherapy)?

To respond to this question, we compared the proportion of Bcl-xL (low) and Bcl-xL (int/high) TANs in KP, SNL and chemotherapy-treated SNL tumors. This indeed reveals a comparable expression of Bcl-xL in naïve KP compared to chemotherapy-treated SNL tumors, which suggests a threshold of Bcl-xL expression above which TANs support tumors in a Bcl-xL dependent manner. We provide this data here below for Reviewer and as new Fig. EV4A. We have added a corresponding text, p.9.

12) In their previous study, the authors show that the use of the anti-neutropenic agent G-CSF potentiates the anti-tumor effect of Bcl-xL inhibition. What would be the effect of G-CSF in this study considering that patients will likely get G-CSF in addition to chemotherapy

We agree that testing the effect of G-CSF is clinically relevant, as patients might suffer from chemotherapy-mediated neutropenia and receive this treatment. However, given that SNL mice did not suffer from chemotherapy-induced neutropenia, we thought the use of G-CSF could be better investigated in a follow-up study, with SNL mice treated for different durations of chemotherapy / G-CSF / Bcl-xL inhibitor, and their combinations. This would require several new mouse cohorts and ethical clearance and would thus significantly extend the duration of the current revision. We therefore hope Reviewer will agree with us incorporating this experimental question into another study.

13) In general, it is unclear why TANs are sometimes classified as SiglecF low vs int vs high and sometimes low vs int/high

We chose to classify TANs into SiglecF low vs. int vs. high when monitoring Bcl-xL expression, as we thought it could be more informative. In the other experiments, we kept the grouping into low vs int/high.

14) The manuscript does not explore how the N1-N6 neutrophil states, described in LUAD, translate into the LUSC context. It is not a problem, but it would be interesting to clarify whether distinct TAN states are similarly conserved in SNL tumors if the authors have the data.

To address this point, we did a bioinformatics analysis of the KP mN1-mN6 neutrophil state-associated gene signatures from (Zillionis et al., Immunity 2019) on the scRNAseq data from SNL TANs and PBNs (Mollaoglu et al., Immunity 2018). This reveals a clear enrichment of mN3 and mN5 in SNL TANs compared to PBNs and, in contrast, an enrichment of mN1 and mN2 in PBNs compared to TANs. However, even if significant, it was harder to conclude for mN4 and for mN6 showing a bi-modal distribution.

According to (Zillionis et al., Immunity 2019), in KPs mN1 and mN2 are 10-fold enriched in healthy lung compared to TANs; mN3 and mN6 are exclusively present in tumor tissue, while mN4 and mN5 are 10-20-fold enriched in tumor tissue. We conclude there is a good similarity but also differences between TAN states in the SNL vs. the KP model.

We add this new data here below and as new Figure EV1C-D. We added new text, p.7.

11th Feb 2026

Dear Prof. Meylan,

Thank you for submitting your revised study. We have now received the reports from the referees who were asked to evaluate your revised manuscript. As you will see below, they are satisfied with the revisions, and I will therefore be able to accept your manuscript once the following minor editorial matters are addressed:

1/ Please address the remaining issue mentioned by referee #1.

2/ Manuscript text:

- Please remove the blue highlights and indicate in track changes mode any new modification.
- "Material and Methods" should be renamed "Methods".
 - o Please provide antibody dilutions
 - o Please indicate the origin of the cells, and whether they were authenticated and tested for mycoplasma contamination.
 - o Please define "Rare tumors with extreme growth rate"
- Data availability: please deposit your sequencing data in a public repository and provide the link in the Data Availability Section.
- Please add The Paper Explained to the manuscript text file.

3/ Figures and Appendix:

- Appendix: The nomenclature should be corrected to "Appendix Figure S1" etc in the appendix file and in the manuscript text. Please add page numbers to the table of contents. Please provide more detailed figure legends for Appendix Figure S1 and S3.
- Callouts: there is a citation for a Table S3 in the text, please correct.

4/ Checklist: please fill in the section "Cell materials/ authentication and mycoplasma contamination".

5/ As part of the EMBO Publications transparent editorial process initiative (see our Editorial at <http://embomolmed.embopress.org/content/2/9/329>), EMBO Molecular Medicine will publish online a Review Process File (RPF) to accompany accepted manuscripts.

We note that you agree with the publication of the RPF, as is.

I look forward to receiving your revised manuscript.

Yours sincerely,

Lise Roth

***** Reviewer's comments *****

Referee #1 (Remarks for Author):

The authors have performed additional experiments and revised the manuscript to address my previous comments. The new data lend additional support to linking chemoresistance to neutrophils, and hence the possibility of targeting TANs as part of a combination therapy to sensitize SNL tumors to chemotherapy.

There is a minor point that requires clarification. What do the several "(" inserted in lines 439, 442, 444 446, 451, 452 and 454 mean? Please correct.

Referee #3 (Comments on Novelty/Model System for Author):

The revised manuscript is now of high quality for publication in embo mol med.

Referee #3 (Remarks for Author):

I would like to thank the authors for answering all the reviewer's comments. I also hope the authors will quickly find a way to resolve their problem with the "brdu" experiments.

***** Reviewer's comments *****

Referee #1 (Remarks for Author):

The authors have performed additional experiments and revised the manuscript to address my previous comments. The new data lend additional support to linking chemoresistance to neutrophils, and hence the possibility of targeting TANs as part of a combination therapy to sensitize SNL tumors to chemotherapy.

There is a minor point that requires clarification. What do the several "(" inserted in lines 439, 442, 444 446, 451, 452 and 454 mean? Please correct.

The parentheses were just a way to reference functions used within Seurat in the writing of methods. We have removed them to avoid any confusion.

Referee #3 (Comments on Novelty/Model System for Author):

The revised manuscript is now of high quality for publication in embo mol med.

Referee #3 (Remarks for Author):

I would like to thank the authors for answering all the reviewer's comments. I also hope the authors will quickly find a way to resolve their problem with the "brdu" experiments.

26th Feb 2026

Dear Prof. Meylan,

Thank you for addressing the last editorial concerns. I am pleased to inform you that your manuscript is accepted for publication and is now being sent to our publisher to be included in the next available issue of EMBO Molecular Medicine.

Your manuscript will be processed for publication by EMBO Press. It will be copy edited and you will receive page proofs prior to publication. Please carefully check the link provided for the data deposited on zenodo.

Please note that you will be contacted by Springer Nature Author Services to complete licensing and payment information. You may qualify for financial assistance for your publication charges - either via a Springer Nature fully open access agreement or an EMBO initiative. Check your eligibility: <https://link.springer.com/journal/44321/how-to-publish-with-us>

With kind regards,

Lise

>>> Please note that it is EMBO Molecular Medicine policy for the transcript of the editorial process (containing referee reports and your response letter) to be published as an online supplement to each paper. If you do NOT want this, you will need to inform the Editorial Office via email immediately. More information is available here: <https://link.springer.com/partners/embo-press/editorial-policies#Peer%20review>